

# Global Climate Modeling with Improved Precipitation Characteristics by Learning Physics (GRIST-MPS v1.0) from Global Storm-Resolving Modeling

Yiming Wang[1,3], Yi Zhang[2,3], Yilun Han[4,5], Wei Xue[1], Yihui Zhou[6], Xiaohan Li[2], Haishan Chen[2]

[1] Department of Computer Science and Technology, Tsinghua University, Beijing, China.
[2] State Key Laboratory of Climate System Prediction and Risk Management and College of Atmospheric Sciences, Nanjing University of Information Science and Technology, Jiangsu, China.
[3] 2035 Future Laboratory, PIESAT Information Technology Co., Ltd., Beijing, China.
[4] Department of Earth System Science, Tsinghua University, Beijing, China.
[5] Scripps Institution of Oceanography, La Jolla, CA, USA.
[6] State Key Laboratory of Severe Weather Meteorological Science and Technology, Chinese Academy of Meteorological Sciences, Beijing, China

*Correspondence to:* Yi Zhang (yizhang@nuist.edu.cn)

**Abstract.** This study develops a machine learning (ML)-based physics parameterization suite trained on 80-day global storm-resolving model (GSRM) simulation data, attempting to replace all conventional physics tendencies in a general circulation model (GCM). Our approach strategically selects key prognostic variables as input features, enabling an effective emulation of multiscale flow interactions of the GSRM by the GCM via dynamics-physics coupling. The resulting ML-enhanced GCM achieves stable Atmospheric Model Intercomparison Project (AMIP)-type simulations over six years, surpassing its conventional counterpart with improved precipitation performance—reducing root-mean-square errors by 8% in boreal summer and 16% in winter, compared to observations. Moreover, the hybrid ML-GCM better captures precipitation frequency–intensity spectra, notably mitigating the overproduction of light tropical rainfall and improving the simulation of moderate rain rates. Sensitivity experiments using different neural network architectures (ResNet, CNN, DNN) demonstrate that all configurations can maintain long-term simulation stability, with ResNet showing superior capability in the simulation accuracy. This work presents a transferable framework that leverages km-scale GSRM data to enhance GCM performance via ML integration, offering a potential route to reduce the gaps between two modeling paradigms.

## 1 Introduction

Weather and climate modeling, which both embodies our understanding of the atmosphere and deepens it, currently operates within two distinct paradigms: (i) highly parameterized general circulation models (GCMs), which are extensively utilized in global climate change research initiatives such as the Coupled Model Intercomparison Project (Eyring et al., 2016); and (ii) explicitly resolved global storm-resolving models (GSRMs) with kilometer-scale resolutions spacing that explicitly resolve convective processes (Satoh et al., 2019). These two modeling paradigms remain operationally decoupled



due to the lack of a unified discretization approach that enables seamless resolution transitions (Yu et al. 2019; Brunet et al.
2023; Miura et al. 2023). A major challenge in bridging this gap lies in the representation of moist physical processes, which
govern scale interactions across different modeling paradigms. GCMs rely on cumulus parameterization schemes that
approximate the bulk effect of interactions between moist convection and large-scale circulation, a well-known source of
climate modeling uncertainties (Arakawa 2004; Lin et al. 2022). GSRMs explicitly resolve the coupling between
atmospheric dynamics and microphysics, and support multiscale flow, hopefully yielding more physically realistic cumulus
convection and multiscale interactions. When incorporated into GCMs, these interactions may replace sub-grid eddy effects
relative to the GCM's grid box, alongside representations of heating and cooling effects due to phase changes, radiative
transfer, and friction.

Machine learning (ML) algorithms have been increasingly applied to facilitate this integration (Schneider et al., 2023;
Eyring et al., 2024), raising the prospect of constructing hybrid ML–physics models (Krasnopolsky and Belochitski 2020).
Ideally, such models would not only perform robustly at a specific resolution but also enable a smooth transition across
multiple meteorologically significant scales, from the typical GCM resolution (100 km) to the GSRM resolution (1 km). The
physical tendencies can be learned separately either to replace an individual scheme (e.g., Chen et al. 2023; Heuer et al. 2024;
Morcrette et al. 2025), or to replace the entire tendency from the physics suite. This study focuses on the latter approach.
Currently, several methods exist for constructing hybrid ML–physics models using this approach. The online learning
strategy, which leverages differentiable numerical solvers to match model outputs with reference/observation datasets, has
demonstrated promise in generating reasonably realistic climate simulations (Kochkov et al. 2024). A challenge lies in
interpreting the nature of the learned physics in this approach. It remains unclear whether the learned tendencies stem purely
from real physical processes (e.g., phase change, eddy effect, friction, radiative heating, etc), or if they also incorporate
certain additional components such as the nudging tendency, which can be independently learned (Bretherton et al. 2022); or
like state correction, which combines conventional numerical models with ML model (Arcomano et al. 2022). The physical
meanings of these tendency terms are different.

Another approach is to directly learn physical tendencies from their generating sources. These sources may include
high-resolution process models (e.g., large-eddy simulations, cloud-resolving models) or observational datasets (Zhu et al.
2022; Bracco et al. 2025). For instance, ML schemes trained on physical tendencies derived from super-parameterized
GCMs (e.g., Rasp et al., 2018; Gentine et al., 2018; Han et al., 2020) have demonstrated the ability to retain the physical
fidelity of super-parameterized modeling while significantly reducing the computational cost. Several operational
implementations of such models have achieved multi-year simulation stability in realistic configurations (e.g., Han et al.
2023; Mooers et al., 2021; Wang et al., 2022). In contrast, GSRMs do not impose artificial scale separation, and learning
physics tendencies from GSRMs presents a unique advantage by allowing for more physically consistent multiscale flow
interactions that closely align with real-world atmosphere. Brenowitz and Bretherton (2018) used neural network-based
parameterizations using coarse-grained GSRM data, demonstrating multi-year simulation stability in low-resolution aqua-
planet scenarios. Yuval and O'Gorman (2020) employed random forests trained on three-dimensional cloud-resolving model





outputs to emulate fine-scale processes in coarse-grid systems. Yuval et al. (2021) refined this approach by leveraging neural networks, achieving comparable predictive accuracy while reducing memory requirements by a factor of 1,900. These

advancements have primarily been tested in idealized aqua-planet configurations, raising critical questions about their applicability to realistic climate modeling. Watt-Meyer et al. (2024) developed a GCM physics parameterization suite trained on coarse-grained GSRM data under realistic surface boundary conditions, enabling stable 35-day simulations while significantly reducing mean-state precipitation and temperature errors. While this approach has not demonstrated very significant advantages in real-world modeling with respect to certain utilitarian metrics (e.g., mean state error), it has the

potential to reconcile scale disparities from a physically orientated training way.

In this study, we develop a ML-based Physics parameterization Suite (MPS) designed to generate temperature and humidity tendencies for GCMs. We propose a refined training strategy. We have experimented with several neural network architectures, including Residual Neural Networks (ResNet), convolutional and deep neural network (CNN, DNN). A sensitivity analysis uncovers that different network architectures produce divergent equilibrium climate states despite

identical training data and hyperparameter configurations are used. The optimal outcome from this work achieved long-term stable Atmospheric Model Intercomparison Project (AMIP)-type climate simulations more than six years, and produces simulation results comparable or better than those produced by a conventional physics suite (CPS).

The remainder of this paper is organized as follows. Section 2 presents the data and methods. Section 3 presents the simulation results and discusses sensitivity of neural networks. Section 4 gives a summary and outlook.

## 2 Model, Data and Methods

### 2.1 Model description and high-resolution GSRM data

The hybrid modeling framework is based on the Global-Regional Integrated Forecast System (GRIST). The features of dynamical core framework of GRIST are detailed in Zhang et al. (2019), Zhang et al. (2020) and Zhang et al. (2024). The baseline physics suite is described in Li et al. (2023), with some improved schemes given by e.g., Li et al. (2022), and Li et

al. (2024). For this study, we adopt the PhysW suite.

GRIST is employed in two configurations: (i) a high-resolution (5 km) GSRM-style setup for generating training data for the MPS, and (ii) a coarse-resolution (120 km) GCM-style setup for applying and evaluating the MPS. Both configurations feature 30 vertical layers. The GSRM setup uses the nonhydrostatic dynamical core with explicit convection (i.e., the cumulus scheme is disabled), following the approach of Zhang et al. (2022). Obviously, the quality of the GSRM

data is critical for the effective development of the MPS. In Zhang et al. (2022), the model successfully captured the multiscale interactions between moist convection and large-scale circulation. Their simulations demonstrated that the time-averaged characteristics of these interactions are comparable to those produced by the GRIST-GCM configuration with conventional cumulus parameterization, but supports better transient features (e.g., extreme rainfall intensity). While the GRIST-GSRM configuration exhibits slightly higher mean-state precipitation biases, it shows superior skill in reducing



systematic errors, such as the excessive frequency of light tropical rainfall. This underscores the importance of replicating the GSRM-resolved multiscale interactions for developing an effective MPS applicable to GCMs.

The GCM configuration follows the setup described in Zhang et al. (2021), using the hydrostatic dynamical core coupled with the conventional PhysW suite (CPS), where the cumulus parameterization is enabled. All other physics schemes—including microphysics, boundary layer, radiation, surface layer, and land surface model—are identical between

the GSRM and GCM configurations, thereby ensuring maximum consistency. Some other details of the two configurations are provided in Table 1.

To enhance the representativeness of the training data, we select four 20-day periods that span different seasons and capture key phases of the El Niño–Southern Oscillation (ENSO) and Madden–Julian Oscillation (MJO), as summarized in Table 2. These periods collectively ensure comprehensive seasonal coverage—January (boreal winter), April (boreal spring),

July (boreal summer), and October (boreal autumn)—and systematically represent the dominant ENSO–MJO interaction regimes that drive climate variability. The current choice of 80 days reflects a practical limitation due to computational and resource constraints, but it already allows essential atmospheric physical processes to be effectively sampled using a limited set of time windows. That said, increasing the number of training samples may further enhance the performance of the MPS.

## 2.2 Coarse graining and data preprocessing

We extract multiscale flow interactions in the GSRM using a thermodynamic framework following Yanai et al. (1973), in which the apparent heat source ($Q_1$) and apparent moisture sink ($Q_2$) serve as mathematical representations of these interactions. These quantities are derived from coarse-grained GSRM data (at 0.25° resolution) using the residual method (e.g., Zhang and Chen 2016), with the governing equations shown in Figure 1 (the middle section of the left panel). Although the present study coarse-grains GSRM data to a fixed resolution, the residual method allows efficient transitions

from arbitrarily high-resolution models to GCM target scales, thereby enabling the MPS to be inherently scale-aware. Establishing a robust physical correspondence between GSRMs and GCMs opens the door to unified simulations of atmospheric processes within a single modeling framework—enhancing both theoretical understanding and predictive skill across multiple timescales. This architecture-agnostic framework offers two advantages: (i) resolution adaptability, preserving essential subgrid-scale variability across different GSRM configurations, and (ii) interoperability with the broader

modeling community using standard atmospheric variables, which are typically available in the standard outputs. Several key design choices are further highlighted below.

**Choice of Large-Scale Variables.** Some preliminary tests identified the optimal set of input features to include temperature ($T$), specific humidity or mixing ratio ($q$), horizontal wind components ($U$ and $V$), and surface pressure ($P$). Although the inclusion of vertical velocity ($\omega$) is theoretically advantageous, it was found to introduce numerical instabilities

in regions with complex topography—a result consistent with previous studies (Clark et al. 2022, Rasp et al. 2018 and Watt-Meyer et al. 2024). All prognostic variables were normalized using min–max scaling, based on their extrema within the 80-day training dataset.



**Vertical coordinate alignment.** In the residual method, advection tendencies are preferably computed on pressure levels to ensure dynamical consistency. However, for machine learning training, it is more desirable to use the model's native hybrid coordinate, which avoids topographic distortion during runtime. Directly computing advection tendencies offline on the hybrid coordinate are inaccurate, as the generalized vertical velocity cannot be reliably estimated from coarse-grained data. To reconcile this discrepancy, we implement a two-step procedure. In Step I, GSRM variables on the hybrid coordinate are interpolated to pressure levels for the sole purpose of computing advection tendencies. In Step II, the resulting advection tendencies are interpolated back to the model's hybrid coordinate, where $Q_1$ and $Q_2$ are then derived. Ultimately, all training inputs ($U, V, T, q, P$) and outputs ($Q_1$ and $Q_2$) are defined on the model's hybrid vertical coordinate, ensuring compatibility with the runtime model structure while preserving physical accuracy in the derivation process.

**Temporal refinement.** To enhance temporal resolution, we applied linear interpolation to convert hourly model outputs into 20-minute interval data, effectively tripling the temporal sampling frequency. This refinement is crucial for improving stability and accuracy of online model integration, as it better aligns the temporal characteristics of the training data with the time step of the target GCM (see Section 2.4).

## 2.3 Training the MPS

The MPS leverages residual neural network architecture by default, with tailored modifications for atmospheric column physics. Central to the design are one-dimensional convolutional layers that explicitly resolve vertical couplings in temperature and humidity profiles, particularly during deep convective events where multi-level interactions dominate subgrid energy transfer. To balance representational capacity with computational efficiency, the network employs five optimized residual units (ResNet5, Figure 1)—a depth empirically determined to preserve most validation accuracy of deeper architectures while saving a lot of training time and resources. Hyperparameter optimization converged on an initial learning rate of $3\times10^{-4}$ with exponential decay (decaying rate=$10^{-6}$ per epoch), minimizing mean absolute error (MAE) across training runs.

To optimize computational efficiency while maintaining global representativeness, we implemented a stratified spatiotemporal sampling strategy. Each temporal snapshot (20-minute interval) extracts 86,400 grid columns distributed across key climate regimes: 50% from tropical latitudes (30°S–30°N) where convective processes dominate, 30% from mid-latitudes (60°S–30°S and 30°N–60°N) capturing baroclinic eddy activity, and 20% from polar regions (90°S–60°S and 60°N–90°N) resolving radiative-polar amplification feedbacks. This geographic weighting generates 497,664,000 training samples (80 days × 24 hours × 3 samples/hour × 86,400 columns). The network underwent 100 training epochs with early stopping (patience=5 epochs, $\delta val\_loss$<0.5%) to ensure full data utilization without overfitting.

Rigorous offline evaluation is important for transitioning ML physics into an operational tool. We quantify emulation fidelity through two complementary metrics: (i) domain-averaged mean squared error (MSE < $1\times10^{-4}$) and (ii) vertical-latitude cross-sections of the coefficient of determination ($R^2$ > 0.3 across most of tropical and midlatitude tropospheric grid points; Figure 2), which collectively verify process-level skill in moisture-convection coupling. Networks satisfying both





thresholds proceeded to online testing. This dual-criterion screening prevents numerically stable but physically implausible models from entering computationally intensive integration phases.

Superior offline performance alone does not guarantee online stability, as the effects of physics-dynamics coupling cannot be purely grasped through offline training. To address this, the shortlist of networks that meet our predefined offline criteria was the first step, then we subject them to online testing. The final selection of our optimal MPS is based on a dual evaluation: satisfying offline performance benchmarks and demonstrating stability in online integration.

## 2.4 Importance of using balanced spatiotemporal sample and temporal resolution alignment

During model development, we identified two key factors that significantly improve the stability and accuracy of the MPS. The first is achieving a more balanced spatiotemporal sample. Initial experiments using the full spatial samples (1440 x 720 grid columns per timestep) combined with coarse temporal sampling (hourly data) led to numerical instabilities during online integration. This instability stemmed from an extreme space–time sampling ratio, which caused the neural network to overfit spatial patterns while failing to adequately learn temporal evolution. To address this issue, we adopted a stratified spatiotemporal subsampling approach: at each timestep, only 86,400 geographically distributed columns were randomly selected, and the temporal resolution was increased to 20-minute intervals via linear interpolation. This strategy balanced spatial and temporal dimensionality while effectively increasing the number of training samples, encouraging the network to focus on both the time evolution of atmospheric processes and static spatial features. This optimized sampling method reduces the training cost and enhances the stability of online integration, highlighting that careful data curation alone, without changes to model architecture, can overcome key challenges in machine learning–physics integration.

The second key aspect is aligning the temporal resolution of the data with the model's integration time step. As noted earlier, we refined the temporal resolution of the large-scale variables by linearly interpolating hourly data to 20-minute intervals prior to computing $Q_1$ and $Q_2$ tendencies. This refinement offers two primary benefits. First, it effectively triples the number of training samples, thereby improving the representation of fast convective adjustment processes that are critical for accurately predicting subgrid tendencies. Second, the use of linear interpolation is justified for large-scale state variables, which typically evolve quasi-linearly over sub-hourly timescales ($\Delta t < 1$ hr). However, this assumption does not hold as well for $Q_1$ and $Q_2$, which exhibit stronger spatiotemporal nonlinearity. As such, performing interpolation only on the input variables—rather than generating full 20-minute GSRM outputs—achieves a 2/3 data compression ratio compared to storing the full-resolution dataset.

The systematic evaluation of training strategies (Table 3) highlights the critical role of spatiotemporal data optimization in governing model performance. In the baseline experiment (EXP1), which employed neither spatial subsampling nor temporal refinement, the model-maintained stability for only three years. Introducing spatial subsampling alone (EXP2) extended stable integration to six years. Further incorporating 20-minute temporal interpolation in EXP3—i.e., full spatiotemporal optimization—maintained six-year stability while substantially reducing the tropical precipitation RMSE by 42% (2.78 mm/day vs. 4.81 mm/day in EXP1). Compared to EXP2, EXP3 yielded a 10% reduction in six-year mean





precipitation RMSE (2.81 mm/day vs. 3.12 mm/day), demonstrating the additive benefit of temporal refinement beyond

spatial subsampling alone.

## 2.5 Online GCM simulation workflow with the MPS

The ML-physics-hybrid GCM builds upon the GRIST framework, with the control experiment (CPS) replicating the configuration described in Zhang et al. (2021) (Table 1, GRIST-CPS). To interface the Fortran-based GRIST model code with the PyTorch-formatted MPS, we implemented bidirectional coupling through the Ftorch library—a framework enabling
real-time tensor exchange between the dynamical core and pretrained neural networks while maintaining operational efficiency.

The online implementation (Figure 1, right panel) adopts a modular architecture, in which the GRIST-GCM dynamical core iteratively transfers atmospheric state tensors to the MPS. The MPS, interfaced via Ftorch, returns $Q_1$ and $Q_2$ tendencies, while legacy CPS diagnostic modules—such as radiation and land-surface coupling—remain unmodified. By restricting
replacements to the physical tendency generation components and preserving the native diagnostic workflow, the framework mirrors the CPS substitutions and ensures full backward compatibility. The replaced CPS components include tendencies from the cumulus parameterization, cloud microphysics, boundary layer scheme, and radiative transfer. The radiation module—the most computationally expensive element in the CPS—is still activated to generate surface fluxes for the land surface and may require a special training in future. The surface layer and land surface models are also retained in their
original form, consistent with standard CPS configurations. Surface precipitation is diagnosed by the MPS via vertically integrated moisture tendency equation, using the relation: $Prec = -\frac{1}{g}\int (Q_2)dp$. Whether to exclude surface evaporation rate is used as an optional procedure for tuning.

Due to the MPS's coarser vertical resolution in the lower troposphere ($\Delta z$ exceeding 200 m below 850 hPa), we retain CPS-derived temperature tendencies ($Q_1$) in the lowest four model levels and moisture tendencies ($Q_2$) in the lowest two
model levels. This selective preservation, validated through sensitivity experiments, serves as a stability-enhancing mechanism. Meanwhile, as in prior studies (Brenowitz and Bretherton, 2019; Clark et al., 2022; Watt-Meyer et al., 2024), we apply vertical truncation of the MPS-predicted $Q_1/Q_2$ tendencies above 300 hPa, effectively excluding the top 12 model layers from machine-learned physics. This hybrid replacement strategy demonstrates that partial physics–ML integration can achieve climate fidelity comparable to a full replacement, while reducing computational costs and mitigating numerical
instability.





## 3 Results

### 3.1 Real-world climate simulations

Two six-year AMIP-style simulations were conducted at 120 km horizontal resolution: a control experiment with the CPS and an ML-enhanced counterpart with the MPS. We evaluate the zonal mean vertical structures of long-term mean

temperature ($T$), specific humidity ($q$), and zonal wind ($U$) represent direct prognostic targets of the MPS (through $Q_1$ and $Q_2$ tendencies), while $U$ emerges as a dynamically constrained diagnostic variable reflecting momentum redistribution. ERA5 reanalysis data (Hersbach et al., 2020) serve as the observational benchmark, with all model outputs regrided to 1°×1° resolution using conservative remapping.

Figure 3 demonstrates close alignment between GRIST-MPS and GRIST-CPS in simulating zonal-mean vertical

structures. Both models exhibit temperature deviations (shading) within ±5 K of ERA5 reanalysis, demonstrating consistent cold biases in the polar lower stratosphere and warm biases in the tropical upper troposphere. Specific humidity profiles (black contours) display nearly identical vertical distributions between configurations. The structure of the zonal wind ($U$) form a wedge-like structure with the humidity, showing little differences in midlatitude jet core positions.

Precipitation is evaluated against the Global Precipitation Measurement (GPM) Product (Huffman et al., 2019). Both

configurations realistically capture the boreal summer (JJA) precipitation dipole—the Intertropical Convergence Zone (ITCZ, 5°N–10°N) and South Pacific Convergence Zone (SPCZ, 5°S–15°S) with maximum rates exceeding 12 mm/day over the Bay of Bengal and western Pacific warm pool (Figures 4a-c). GRIST-MPS reduces global precipitation RMSE by 8% (3.46 mm/day versus 3.75 mm/day) relative to GRIST-CPS, primarily through improved ITCZ localization: the MPS better constrains the ITCZ core position relative to GPM observations, whereas the CPS produces a wider rainfall band.

Extratropical performance remains comparable, with both models capturing most of observed midlatitude storm-track variance (55°N–65°N). The MPS slightly underestimates precipitation over southern oceans (30°S–50°S), while the CPS shows some overestimations extending to 70°S.

During boreal winter (DJF), GPM observations reveal a meridionally contracted state of tropical rainbands and intensified midlatitude storm-track precipitation (45°N–60°N, Figure 4d). Both configurations capture this seasonal

transition (Figure 4e, f), with GRIST-MPS demonstrating enhanced fidelity through a 16% RMSE reduction (3.16 mm/day versus 3.76 mm/day) achieved by narrowing the width of ITCZ rain band.

Meanwhile, residual biases persist in GRIST-MPS: a 15%–20% overestimation of summer tropical Indian ocean rainfall (10°S–10°N, 65°E–95°E) and a systematic 1-3 mm/day underestimation of Southern Ocean (50°S–60°S) and Maritime Continent (5°S–5°N, 95°E–150°E) precipitation across seasons.

Both configurations accurately reproduce the observed seasonal migration of tropical precipitation maxima (Figure 5), with boreal summer peaks centered near 10°N aligned with the northward-migrating ITCZ. However, systematic discrepancies emerge in the meridional range of precipitation representation: GRIST-CPS overestimates the ITCZ's meridional extent, generating broadened rainfall distributions characteristic of overactive convective initiation in cumulus



parameterizations. GRIST-MPS demonstrates superior constraint of ITCZ width but exhibits a little underestimation over 0-
10°N during July–September.

To systematically assess the performance of GRIST-MPS in characterizing complex atmospheric systems, we employ
the East Asian Monsoon as our case study. Our analysis utilizes established East Asian monsoon index (EAMI) from prior
studies as benchmark metrics (Zhu et al. 2005). The EAMI takes the influence of the annual cycle of the meridional and
zonal sea-land thermal differences into account in the East Asia-Pacific region and reasonably describes the characteristics of
the annual cycle of the transition between the East Asian winter and summer monsoons, which is defined as:

$$
\begin{aligned}
EAMI = (U_{850hPa} - U_{200hPa}) *_{(100-130°E, 0-10°N)} \\
+ (SLP_{160°E} - SLP_{110°E}) *_{(10-50°N)}
\end{aligned}
\tag{1}
$$

where $U$ represents area-averaged $(100 - 130°E, 0 - 10°N)$ monthly mean zonal winds (dimensionless), $SLP$ denotes
averaged monthly sea level pressure ($10 - 50°N$) (dimensionless), and the asterisk (*) operator indicates variable
standardization through mean removal and unit-variance scaling ($X = (X - \mu)/\sigma$). This enables a quantitative assessment
of the model's ability to capture both the seasonal evolution and interannual variability characteristics of monsoon dynamics.

We computed the East Asian Monsoon Index (EAMI) for monthly variables and derived its climatological seasonal
cycle across a six-year period (Figure 6). Both GRIST-CPS and GRIST-MPS successfully replicate the observed seasonal
monsoon phase, capturing the July maximum and February minimum. While GRIST-CPS simulations align closely with
observations, GRIST-MPS exhibits a systematic bias: it overestimates monsoon intensity prior to July and underestimates it
post-July. This indicates that GRIST-MPS could simulate the annual cycle of the East Asian monsoon, even though the
training data only includes 80 days. This outcome strongly motivates a further refinement of MPS for extended climate
applications.

To more comprehensively reveal the ability to simulate precipitations of the two configurations, we analyze tropical
precipitation frequency distributions (30°S–30°N; 2001–2006). Precipitation is classified into four intensity categories: light
(0.1–10 mm/day), moderate (10–25 mm/day), heavy (25–50 mm/day), and extreme (>50 mm/day). Besides GPM
observations, the ensemble means values of 11 CMIP6 models (CESM2, CESM2-WACCM, CMCC-CM2-SR5, E3SM-2-0,
E3SM-2-0-NARRM, EC-Earth3, EC-Earth3-AerChem, GFDL-CM4, MRI-ESM2-0, SAM0-UNICON, TaiESM1; hereafter
CMIP6-ENS) are included. In relative to GPM data, both CMIP6-ENS and GRIST-CPS overestimate total precipitation
occurrence by 45% and 51%, respectively (Figure 7)—consistent with earlier documented biases (Fu et al., 2024). The MPS
reduces this discrepancy to 22%. Specifically, it reduces light and moderate rain overprediction by 23% and 16%,
respectively, while preserving observed heavy/extreme precipitation frequencies. This demonstrates that MPS effectively
mitigates persistent precipitation distribution errors without compromising extreme event statistics.

## 3.2 A sensitivity analysis of different neural networks

Besides ResNet, we have also integrated two alternative neural network architectures—Convolutional Neural Networks
(CNN) and Deep Neural Networks (DNN)—to examine the sensitivity of online simulations to neural networks. The three





networks are trained by identical datasets and preprocessing procedures. The switch of each network during the GRIST-MPS runtime only needs to change the weight file.

Comparative analysis of neural architecture reveals distinct thermodynamic fidelity characteristics (Figure 8). ResNet architecture demonstrates superior temperature profile reconstruction, maintaining deviations < 5 K from ERA5 reanalysis
throughout the troposphere. In contrast, CNN and DNN architectures exhibit systematic warm biases (5-10 K) between 300–600 hPa, while DNN exhibit warm biases at both North and South pole. Humidity simulations further highlight architectural divergence: while CNN/DNN architectures compress moisture profiles toward lower altitudes (peaking at 850 hPa with about 50% faster moisture decay rates above 500 hPa), ResNet and DNN preserves physically consistent specific humidity gradients up to 300 hPa, a capability enabling enhanced representation of upper-tropospheric moist processes. Wind field
simulations demonstrate architectural invariance, indicating dynamical core constraints predominantly govern momentum balance regardless of physics parameterization. These findings indicate that neural network selection significantly influences thermodynamic fidelity which is a critical design consideration for developing ML-based parameterizations.

Neural architecture selection induces substantial discrepancy in precipitation simulations, particularly in tropical convective organization (Figure 9). During boreal summer, the CNN architecture overestimates western Pacific and tropical
Indian Ocean precipitation relative to observations, generating an excessively broad ITCZ with spurious drizzle artifacts across subtropical highs. The DNN exhibits systematic 15%–20% underestimation globally while maintaining comparable spatial RMSE to observations (3.08 versus CNN's 3.67 mm/day).

Winter simulations of CNN reveal pronounced biases: precipitation over SPCZ exhibits large (about 20%) overestimation relative to observations. The DNN's underestimation persists at 8%–10% magnitude but shows improved
spatial pattern alignment with ResNet. The ResNet architecture consistently outperforms other configurations in maintaining a small deviation across seasons. These systematic discrepancies highlight how architectural inductive biases—specifically, the CNN's excessive sensitivity to localized features compared to the DNN's global feature integration—substantially influence precipitation simulations. This underscores the critical need for architecture-specific uncertainty quantification in machine learning-driven climate modelling, as model design disparities directly shape predictive outcomes.

Seasonal precipitation migration patterns reveal distinct architectural sensitivities (Figure 10). While all architectures capture fundamental north–south displacement of tropical precipitation maxima, CNN simulations exhibit 30% greater meridional spread, consistent with documented overestimation of tropical precipitation (Figure9 b, e). Conversely, DNN systematically underestimates peak monsoon intensities by 38%, a deficiency attributable to its limited capacity in resolving nonlinear moisture-convection feedback inherent to fully connected architectures. ResNet maintains the closest fidelity to
observed seasonal progression (<5% phase error in ITCZ migration timing).

The frequency-intensity spectra of precipitation (Figure 11) reveal neural architectural influences on precipitation distribution characteristics: CNN amplifies nearly double of the conventional GCM bias through over simulation of light precipitation. Conversely, the DNN achieves the closest alignment with observed frequency distributions despite systematically underestimating heavy precipitation (>50 mm/day). This apparent paradox originates from DNN's inherent





regularization properties, its fully connected architecture preferentially attenuates extreme convective events while better constraining pervasive light precipitation (1–10 mm/day) that dominates tropical rainfall occurrence (accounting for >78% of events). ResNet demonstrates intermediate performance, replicating the spectra of CPS (Figure 6: pink/blue curves).

## 4 Summary and outlook

This study establishes a new ML-physics hybrid modeling framework through seamless integration of neural networks
trained on high-resolution GSRM data into the GCM model, achieving stable six-year climate simulations with enhanced process-level fidelity. The major conclusions are given below.

**Major achievement.** The GRIST-MPS exhibits strong thermodynamic consistency, closely replicating ERA5 vertical profiles of temperature ($T$ bias < 5 K) and specific humidity ($q$ bias < 1.5 g/kg), while reducing tropical precipitation RMSE by 8% in boreal summer and up to 16% in boreal winter—primarily through improved representation of convective–diabatic
processes. Key improvements include more accurate ITCZ positioning, phase-aligned midlatitude storm tracks, and enhanced precipitation frequency-intensity spectra, particularly the improved light to moderate range (0.1–25 mm/day). Crucially, the framework maintains long-term numerical stability and accuracy via architectural innovations and optimized spatiotemporal data sampling. These results demonstrate that ML–physics integration can overcome long-standing trade-offs in traditional parameterizations, offering a transformative pathway for next-generation climate modeling. Leveraging
GSRM-driven learning to construct ML–physics hybrid GCMs offers distinct advantages: GSRMs inherently capture multiscale atmospheric interactions without imposing artificial scale separation, while allowing flexible resolution specifications—essential for developing scale-aware parameterization schemes. Furthermore, community-standardized GSRM datasets based on common state variables promote reproducibility and interoperability. We contend that this modeling paradigm paves the way toward unifying GSRM and GCM scales by harnessing the synergy of ML and high-
fidelity data, offering a scalable and physically grounded foundation for future Earth system modeling.

**Remaining challenges.** The current training is limited to an *80-day only* GSRM dataset, future extensions are expected to enhance model generalization and fidelity. One limitation of the present framework is the absence of momentum feedback in the ML architecture, which may lead to systematic biases in upper-tropospheric jet stream positioning (e.g., $U$ bias > 5 m/s at 200 hPa). Additionally, raw GSRM-derived multiscale interactions may require constraints, to fully align with GCM-scale
applications. Despite these limitations, our results demonstrate that GSRM-trained ML–physics suites can achieve simulation stability (over six years) and high physical fidelity (e.g., ITCZ positional refinement within 1° latitude). This may establish a strong foundation for scalable and physically consistent next-generation climate modeling paradigms.

**Further implications.** The ML–physics model introduces a novel computational framework that has interdisciplinary implications. The MPS relies heavily on matrix multiplication, a computational pattern well-suited for optimization techniques (e.g., reduced precision) that align with recent advances in high-performance computing (Chen et al. 2024). In
terms of computational efficiency, the current unoptimized GRIST-MPS shows limited advantage over GRIST-CPS,



primarily due to the activation of diagnostic modules (which can be optimized), and lower-resolution CPS does not present significant overhead. However, targeted optimizations reveal its inherent scalability advantages on the new Sunway architecture: Duan et al. (2025) successfully deployed an earlier version of the MPS suite on the new Sunway supercomputer, significantly accelerating global 1km GRIST-GSRM. This demonstrates that while the baseline MPS performance is constrained by auxiliary computational overhead, its architectural design enables superior acceleration potential when leveraging platform-specific optimizations. Finally, the software framework presented in this study can serve as a general platform for testing AI-trained physics suites within hybrid AI-Physics GCMs.

### Acknowledgments

This research is supported by the National Youth Talent Project (grant no. 2021) and the Startup Foundation for Introducing Talent of NUIST. Editors and reviewers are thanked for their comments and handling of this paper.

### Code and Data availability

Frozen model code, including the MPS, a manual, configuration files and input data, training and plotting scripts and plotting data are available at https://doi.org/10.5281/zenodo.15853268 (GRIST-Dev, 2025). GPM data may be downloaded at: https://gpm.nasa.gov/data/directory. ERA5 data may be downloaded at: https://www.ecmwf.int/en/forecasts/dataset/ecmwf-reanalysis-v5.

### Author contribution

YW performed ML training and online model integration, with inputs from all authors. All the authors discussed this work and contributed to the final manuscript version.

### Competing interests

None.

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





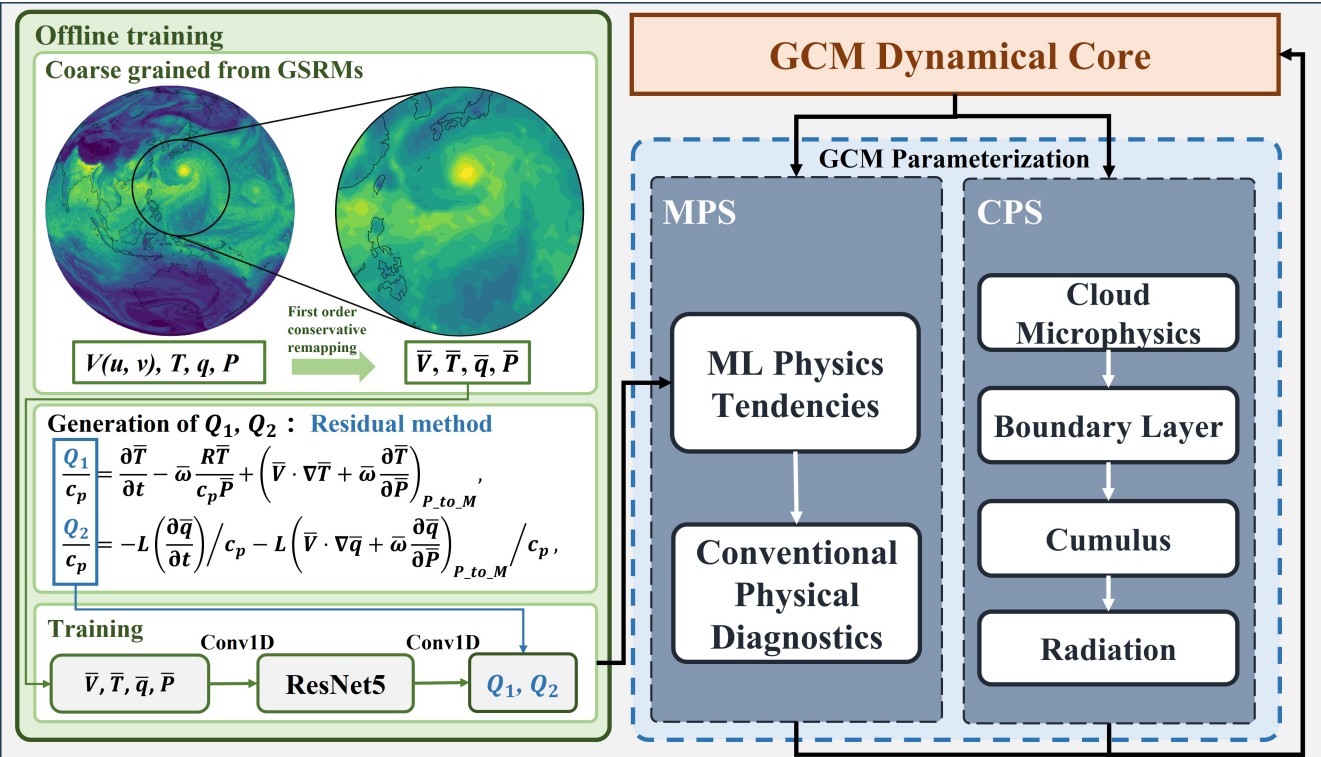

**Figure 1. The workflow of offline training of MPS (Machine-Learning Physics Suite; Section 2.1-2.3) and online simulation of the GCM with ML-physics (Section 2.5). In the equations, $T$ represents temperature, $q$ specific humidity, $V$ horizontal wind components (zonal $u$ and meridional $v$), $\omega$ vertical velocity, $R$ the gas constant for dry air, $P$ the atmospheric pressure at all vertical levels, $c_p$ the specific heat at constant pressure, and $L$ latent heat of evaporation or condensation. The notation $\overline{(\cdot)}$ represents the horizontally coarse graining operator, from 5 km to 30 km in this study. The subscript $P\_to\_M$ represents the conversion from pressure coordinate to the model level, after the calculation of advection terms on the pressure level.**





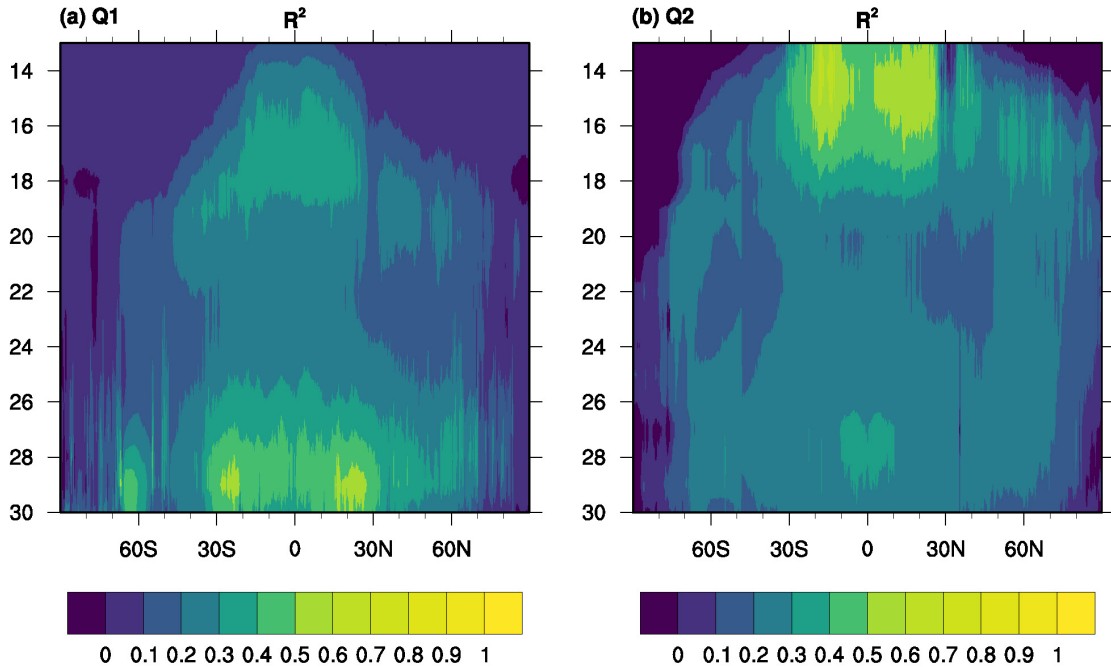

**Figure 2.** Offline skill of the coefficient of determination ($R^2$) for $Q_1$ and $Q_2$, as functions of latitude and model level.



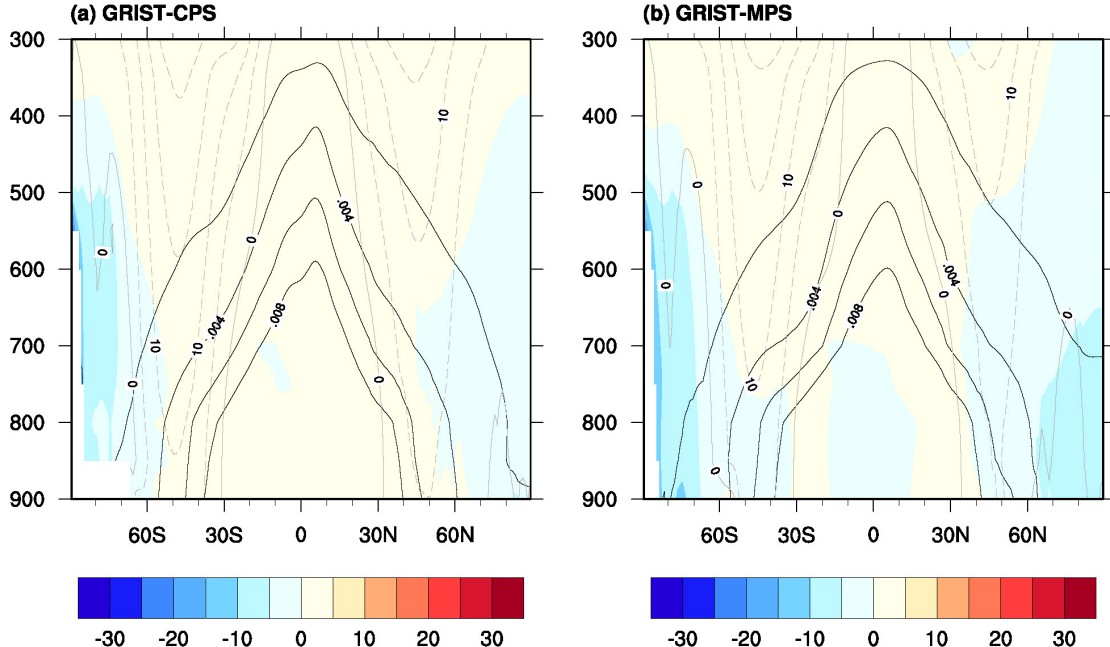

**Figure 3. (a) Latitude–pressure cross section of the time averaged zonal mean temperature differences (contour), climatology specific humidity (black lines) and climatology zonal winds (gray lines) with GRIST-CPS. (b) as in (a) but for GRIST-MPS simulation. The simulation period for all of the models was from 2001 to 2006.**





**Figure 4. The mean precipitation rate (mm/day) averaged from 2001 to 2006 for June–July–August (a, b, c) and December–January–February (d, e, f)  by (a, d) GPM, (b, e) GRIST-CPS, and (c, f) GRIST-MPS.**





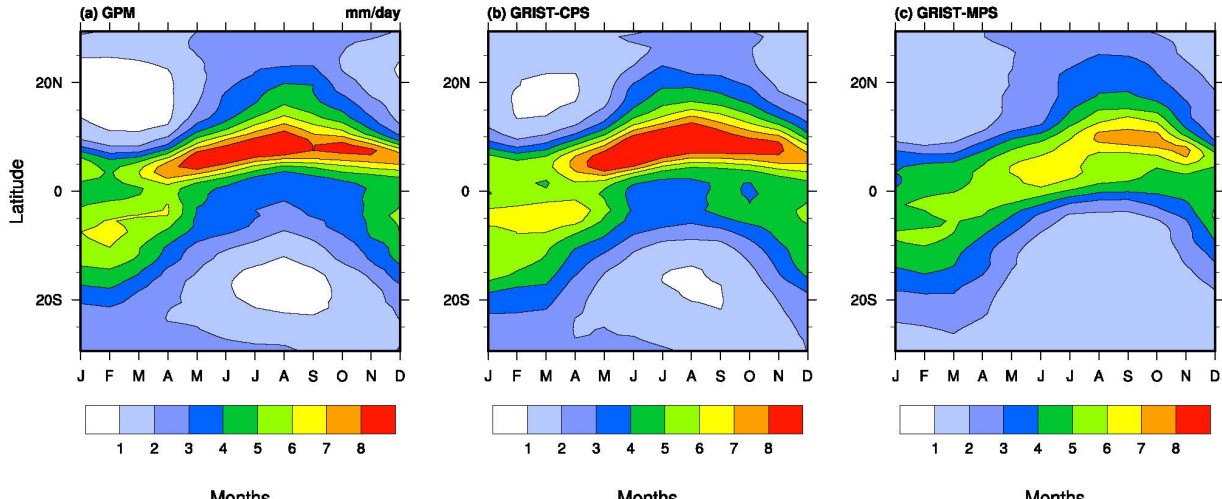

**Figure 5. Seasonal evolution of tropical precipitation from 2001-2009 for observation from (a) GPM, (b) GRIST-CPS, and (c) GRIST-MPS (units: mm/day).**



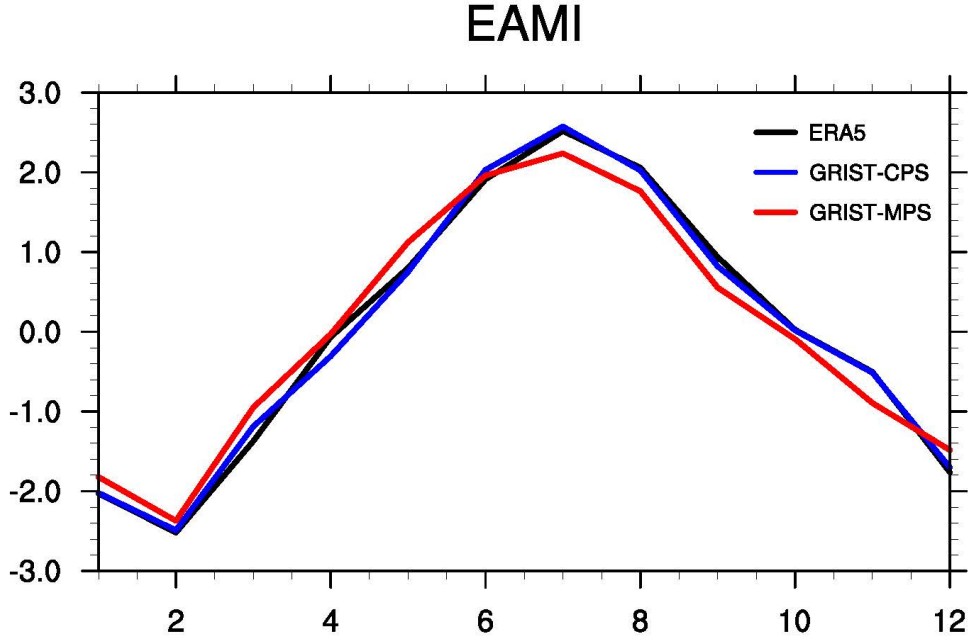


**Figure 6: The East Asian Monsoon Index (EAMI) of GPM (black line), GRIST-CPS (blue line) and GRIST-MPS (red line).**





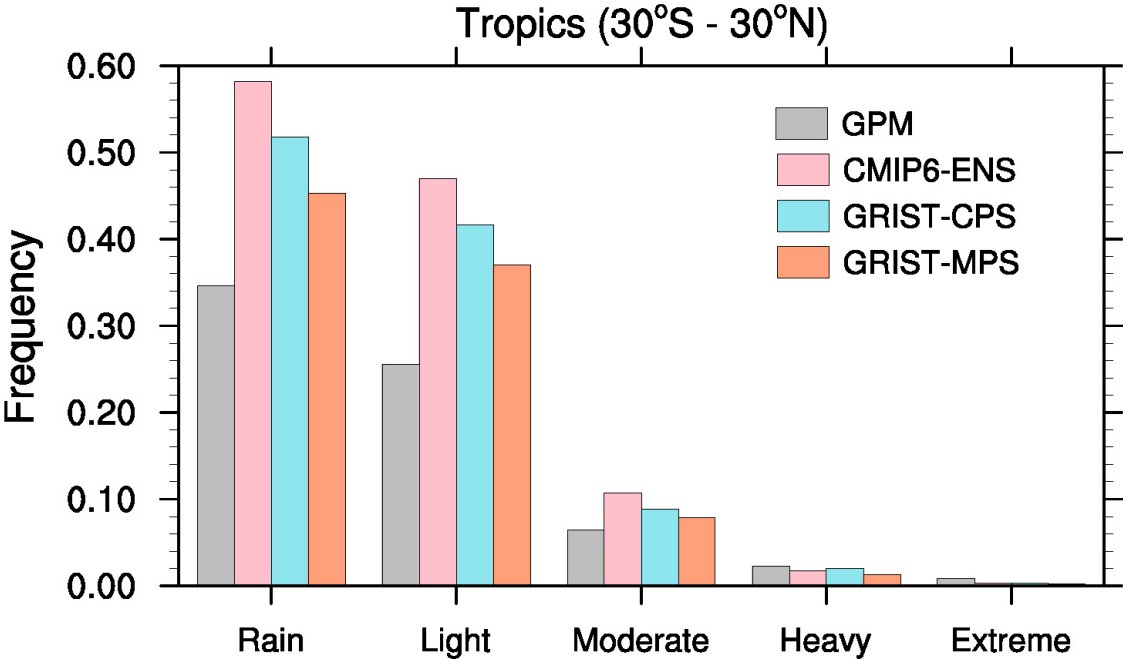

**Figure 7. The frequency probability distributions of tropical daily precipitation from 2001-2006 obtained from GPM (gray boxes), 11 CMIP6 models ensemble mean (CMIP6-ENS; pink boxes), GRIST-CPS (blue boxes) and GRIST-MPS (orange boxes).**



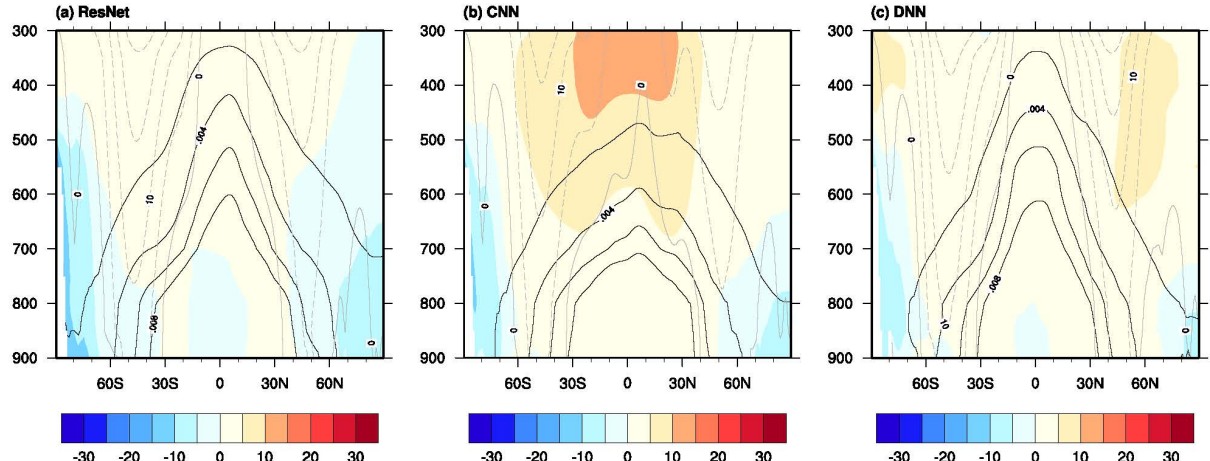

**Figure 8. As in Figure 3 but for (a) ResNet, (b) CNN and (c) DNN.**





**Figure 9. As in Figure 3, but for (a) ResNet, (b) CNN, (c) DNN in JJA, (d)-(f) in DJF.**





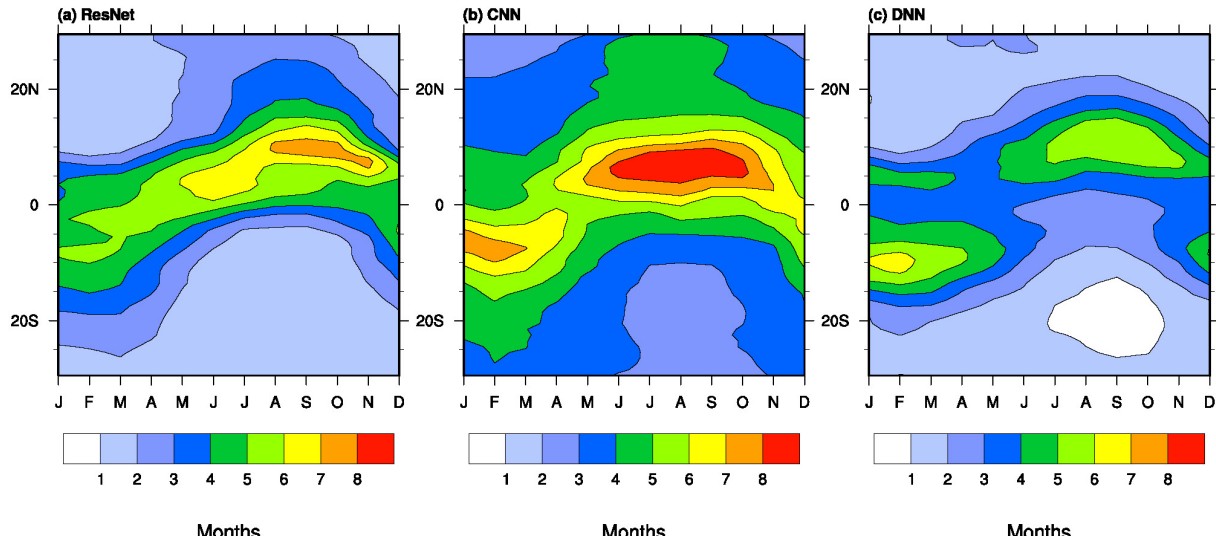

**Figure 10. Same as in Figure 5, but for (a) ResNet, (b) CNN, (c) DNN.**




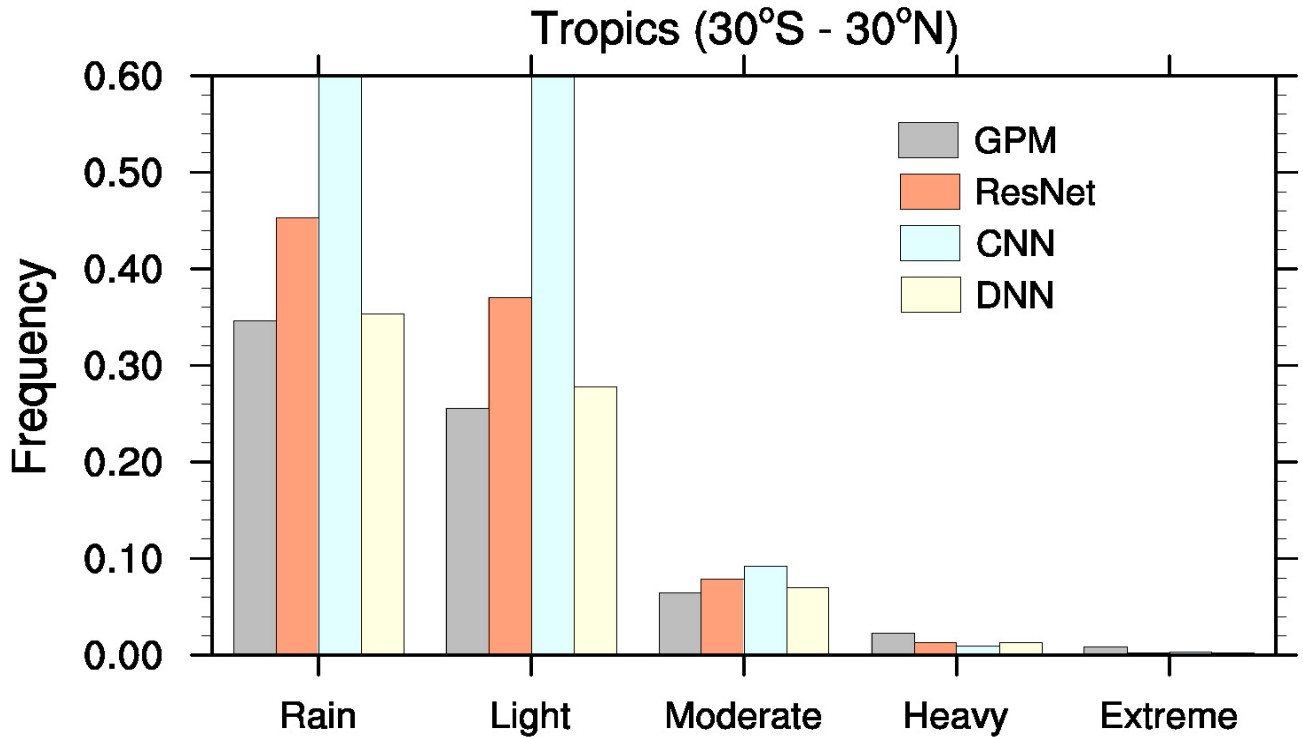

Figure 11. As in Figure 7 but with CNN (yellow bins) and DNN (blue bins) added.




**Table 1 The GSRM and GCM configurations of GRIST for this study.**

| Set up | Dynamics | Horizontal resolution | Dycore/Tracer/Fast Physics time steps(s) | Square of Smagorinsky Coefficient($C_s^2$) | Hyperdiffusion coefficient ($m^4 / s$) |
|---|---|---|---|---|---|
| GSRM | Nonhydrostatic | G9B3(5km) | 6/30/60 | 0.005 | $1\times10^{10}$ |
| GCM | Hydrostatic | G6(120km) | 300/600/1200 | 0.015 | $2\times10^{14}$ |

**Table 2 Selected time periods and climate characteristics.**

| Experiments | Time period | Oceanic Niño Index | Real-time Multivariate MJO index |
|---|---|---|---|
| 1 | 1-20, Jan, 1998 | 2.2(El Niño) | 0.69 to 1.98 |
| 2 | 1-20, Apr, 2005 | 0.4(neutral) | 2.72 to 3.71 |
| 3 | 10-29, Jul, 2015 | -0.4(neutral) | 0.17 to 1.05 |
| 4 | 1-20, Oct, 1988 | -1.5(La Niña) | 0.67 to 2.98 |


**Table 3 The optimal MPS experimental results of each setup.**

| Experiments | Random points selection | Linear time interpolation | Running time | The RMSE of time-averaged precipitation |
|---|---|---|---|---|
| EXP1 | × | × | 3yr | 4.81 |
| EXP2 | √ | × | **6yr** | 3.12(3yr)/3.12(6yr) |
| EXP3 | √ | √ | **6yr** | **2.78**(3yr)/**2.81**(6yr) |