# Peer review of "Global Climate Modeling with Improved Precipitation Characteristics by Learning Physics (GRIST-MPS v1.0) from Global Storm-Resolving Modeling"

_EGUsphere, 2025_

## Referee Comment (RC1)

**Review of Global Climate Modeling with Improved Precipitation Characteristics by Learning Physics (GRIST-MPS v1.0) from Global Storm-Resolving Modeling**

The model configuration proposed by the authors is sound and worthwhile to pursue, with a high quality training data source. Specifically, it would be a highly impactful and somewhat surprising result to demonstrate that MPS can be trained to produce a skillful online parameterization using only 80 years of GSRM training data, especially one using deep learning approaches which require many independent samples to train without overfitting.

However, there are presently critical issues in the evaluation of the methodology that make it unclear the methodology has produced a skillful model, or the extent to which any model skill may be a consequence of p-hacking (due to a lack of detail about model selection strategies).

In my review I have provided specific examples of how the evaluation could be improved to demonstrate the model has skill. The authors should also make it clearer that the trained model was not selected based on the measures shown by explaining exactly how it was selected. I strongly suggest the authors use a validation dataset from the GSRM congruent with the training data to verify the model is not overfit.

Abstract:
- The abstract is light on some specific details which would be valuable to a reader in understanding the contribution of the paper. While these may not be strictly required, adding them would significantly improve communication of the results.
- Should mention the resolution of the GCM/ML, and that the GSRM simulation data is coarsened (coarsening is itself a difficult problem you are tackling here).
- (Optional) It is also worth highlighting the use of full topography.
- (Optional) What is the GCM used?
- (Optional) Says comparison is being done against observations, it would be worth spelling out precipitation observations and historical reanalysis/ERA5. Currently reanalysis comparison is not mentioned in the abstract.

Overall comments:
- To evaluate the quality of the learning, the method output must be compared against the GSRM configuration of the model, which is congruent with the training procedure. Observations and long integrations may also be compared against, but it needs to be clear the

extent to which observations are being matched by properly learning the training data, as opposed to just having less variability. This can only be done to the extent that the match of the GSRM with observations or reanalysis is known/presented. For example, does the GSRM have reduced precipitation compared to the CPS configuration? Does the mean precipitation of the long-term MPS configuration agree with the precipitation present in the GSRM training set? One would imagine it is not biased in the first few timesteps - does the precipitation produced by the MPS configuration stay consistent throughout the run, or does it reduce as the simulation leaves the initial condition?

– Consider including a baseline similar to MPS but where the ML-based predictions of Q1 and Q2 are uniformly zero. At the least, the MPS model should better match the coarsened GSRM data than this baseline, and this comparison shows whether the MPS has any positive skill when integrated online (whether or not it outperforms CPS).

– The word "validation" only appears once in the manuscript, and it is not clear what validation dataset if any are used to measure overfitting and skill during training and hyperparameter optimization. On L91-92, it is implied that GSRM-style data is only used for training, and not for evaluation, making it hard to understand how out-of-sample performance on the loss could be evaluated. I am well aware of the high cost of GSRM output, but at least a few days of independent simulation data should be used for this purpose. Especially given the possibility of overfitting on a dataset only 80 days long, it is crucial to include validation metrics during training.

– The implication of the lack of validation dataset is that the authors tuned the model on the measures being presented in the paper. If this is true, it significantly increases the potential any model skill is due to random chance, especially given that the final model was selected by testing an unnamed number of models.

Line:
- A few lines talk about scale-invariance, though none is shown here. L45: "Ideally, such models would not only perform robustly at a specific resolution but also enable a smooth transition across multiple meteorologically significant scales, from the typical GCM resolution (100 km) to the GSRM resolution (1 km)" and L119 "Although the present study coarse-grains GSRM data to a fixed resolution, the residual method allows efficient transitions 120 from arbitrarily high-resolution models to GCM target scales, thereby enabling the MPS to be inherently scale-aware." It is not clear to me here or from referring to Zhang and Chen 2016 how this is the case. For instance, a model which learns residuals at

a given resolution cannot be applied zero-shot at other resolutions, any more than a model that learns the full field. The model does not appear to be scale-aware as mentioned on L120. I would suggest removing these references to smooth scale transitions, as they are not important to the primary contribution of this paper. If they are kept, they should be supported at least with a theoretical basis, if not experimental results. Note that the results here do successfully lead a coarser model to behave more like a finer model, but this is not what is generally meant by scale-awareness or by a smooth model transition across scales.

- There are some conflicting statements about the configuration of the GCM - it's an important detail whether it's using explicit convection or a cumulus scheme. On L93-94 it says the cumulus scheme is disabled, but on L94 the GSRM is compared against the GCM with a conventional cumulus parameterization. On L103 it mentions the cumulus parameterization being turned on. Any results of MPS should be compared in skill against the optimal GCM configuration for the base model, in this case it seems one with the conventional cumulus parameterization turned on, even if MPS is hooked into a version of the GCM without cumulus parameterization. I think that's what you're doing, but perhaps L93-94 should be modified.
- L117-118: How are Q1 and Q2 computed numerically? For example, are the terms computed by taking gradients of the coarsened fields according to a given numerical scheme (if so, something like "where gradients are computed using [...]"), or by taking the tendency of the dynamical core in coarse mode?
- L142: Please clarify whether model outputs are referring to GSRM values or ML outputs.
- "column" only appears as of line 146, and I only realized this is a column model by line 156. The model being column-local is noteworthy and should be mentioned much earlier.
- L152: What optimizer is used?
- L152: How was model skill evaluated when performing hyperparameter optimization?
- L153: The stated decay rate of 1e-6 per epoch is either incredibly large (if directly multiplicative, giving a final epoch LR of (3e-4)*(1e-600), or incredibly small (if the multiplier is (1 - rate), giving a final epoch LR of 2.9997e-4). Is this accurate?
- L153-154: it is conventional to say you are using a MAE loss function, and how the values are normalized when computing the loss (in this case, it would appear to be using the min-max scaling mentioned elsewhere). It could be worth remarking on why you chose MAE loss over MSE, if there is a reason.

- L160: Please mention the batch size here alongside the number of samples per epoch.
- L162-172: You mention a process of training many models and evaluating for the best candidates. Please specify how many models were trained for this selection process.
- L171: Please specify exactly what is meant by "demonstrating stability in online integration". For example, was there a pass/fail measure of stability? Does the paper present the first trained model for which this stability measure was passed?
- L173-183: No edit needed here, but this result is quite curious and I am skeptical of the interpretation, since the linear interpolation doesn't grant any true additional temporal resolution, and emulator models successfully operate at 6-hour timesteps. It may be that the model behaves better online when it must make 3 of roughly the same (residual) prediction, and its errors get averaged or the intermediate values serve as a type of data augmentation. You might consider for your own purposes training this model with a 1-hour timestep but linearly interpolating the training data from 3-hour intervals down to 1-hour, and see if you get a similarly stable model. This would prove out that it is in fact the 20-minute temporal resolution which is key, and not the structure of linearly-interpolated data.
- L186-188: It is not right to say the effective number of training samples is tripled, given these are not independent samples but rather completely linearly dependent on existing training samples. Rather, this is similar to other regularization strategies used for improving stability, e.g. https://arxiv.org/abs/1912.02781 or Bishop (1995) "Training with Noise is Equivalent to Tikhonov Regularization"
- L215-216: It is somewhat problematic to compute precipitation in this way, as the value is really precipitation minus evaporation. This significantly affects the headline result of the paper (improved preciptitation skill through lower RMSE). It would be appropriate to compare against precipitation minus evaporation for the CPS configuration, and to appropriately describe these values in later sections. For example, this significantly affects the interpretation of Figure 7.
- L223-225: Please quantify the reduction in computational cost of GRIST-MPS (using CPS for the layers described) compared with GRIST-CPS.
- L228: What years are used for the AMIP-style simulations? Are these related to the years used for the GSRM training simulations?
- L243-247: Please comment on the relative strength of the ITCZ between the models, and quantify the underestimation and overestimation of precipitation in the southern oceans.

- L249-251: Can you give evidence this reduced RMSE is due to improved placement of the ITCZ and not reduced strength, for example by using a measure that accounts for the variance of the model?
- Figure 4: GRIST-MPS seems to have significantly reduced mean precipitation, but it is hard to tell from color maps alone. Please include a measure of mean bias, as well as a measure of skill which does not improve for models with less variability, such as the Pearson correlation coefficient. This would make it clear the MPS model is outperforming the CPS model.
- Figure 5: The rainbow color bar is not doing you any favors here, making it look like you have a drastic reduction in precipitation. Using a more linear color bar would give a more accurate representation of the data.
- L259-260: Does this really show superior constraint of the ITCZ width with MPS? The green contour lines appear similar to CPS, and past that the ITCZ appears weakened more than narrowed. The lack of a maximum around -15S Jan-March for MPS is also notable, weakened variabillity in this region is also apparent in Figure 4.
- Figure 7: Please include the coarsened GRIST data used as target for the MPS model. It is not clear whether the MPS is doing "better" because it is accurately representing reduced precipitation as seen in the GSRM, or because it has weakened variability in general compared to CPS.
- L309-310: Please quantify improved spatial pattern alignment. RMSE is not sufficient for this.

---

## Referee Comment (RC2)

**Review of the preprint entitled "Global Climate Modeling with Improved Precipitation Characteristics by Learning Physics (GRIST-MPS v1.0) from Global Storm-Resolving Modeling" by Y. Wang et al.**

**General comments**

This preprint describes the improved long-term global climate simulations using a general circulation dynamic core coupled with machine-learning (ML)-based physics instead of the conventional parameterization schemes. While economic ML large models have been long doubted for lacking of physical constraints, the dynamical models suffer from expensive computational cost with conventional parameterization or unstable computation with online ML physics schemes (MPS). The present study shows a practical way of coupling the ML-physical processes for stable model integration by selecting balanced spatiotemporal resolution of the training data, besides the evaluation of three neural network architectures. The proposed method effectively ameliorates the precipitation simulation and structure of thermodynamic quantities (T and q) via the improved convective-diabatic interaction, even though no physical feedback to momentum is considered. Descriptions of the methodology and corresponding consummate skill in the MPS implementation, which must be interested and also helpful to researchers in this field, are provided in detail for reference.

The use of high-resolution model output as the training data, being of a result of multi-scale dynamical interaction, helps the neural network to learn detailed spatial structure and temporal variation of the corresponding variables from the fine-modelled data. The induced MPS is then capable of reproducing the high-resolution-model pattern of precipitation and prognostic variables with coarse-resolution GCM, just as show in this preprint.

Beside the quality improvement, the computational cost is also an issue of concern. It is better to add a short note on the cost of MPS vs CPS. In this study, the hybrid coupling of CPS and MPS to dynamical core, in addition to retaining of radiation scheme for the land surface model, seems introduce excessive computation. How about the cost reduction with the MPS?

Another concern issue is the diagnosis of the Q1 and Q2 on pressure levels. The interpolation between the model levels and the pressure levels introduces errors inevitably. Why not diagnose directly on the model levels?

The preprint is generally well written and organized. The technical details must be highly evaluated for the online stable integration. I recommend accept for publication after necessary revisions.

**Specific comments**
(1) The GRIST model should be mentioned in the abstract.
(2) Please confirm the "precipitation frequency-intensity spectra" in Line 23, may be "precipitation frequency" because no frequency-intensity spectra is shown in the preprint.
(3) Please rephrase the "explicitly resolved global storm-resolving models" in Line

32.

(4) Line 90, please note the abbreviation "PhysW" here.

(5) (Line 120) In case of a different resolution, the use of MPS needs additional training or not? For the specific resolution, the trained MPS may resolve interactions between systems above the doubled grid scale. For finer resolution, however, the trained MPS may miss the processes smaller than training data resolution (0.25 degree). So, is it really scale-aware?

(6) Line 124, is the "GSRM" a typo of "GCM"?

(7) Why not directly output (U, V, T, q, P) data in a 20-minute interval in the GSRM simulation (Lines 142 & 179)? Only due to the limitation of data storage?

(8) Please check the formula of *Prec* in Line 216. What unit is used here?

(9) Please define X, μ and σ in Line 269.

(10) The word "extreme event" is better replaced with "heavy-precipitation event" for easy understanding. The extreme event, out of the training data, is difficult for MPS to resolve.

(11) What is the "weight file" in Line 292.

(12) The sentence on Line 322 needs to be revised. Do you mean: CNN nearly doubled the frequency of light precipitation occurrence in conventional GCMs?

(13) Can the GRIST-MPS stably run more than 6 years?

(14) It is helpful to label the averaged pressure in right side of panels in Fig.2

(15) In Fig.3, temperature difference should be the shaded (being wrongly noted in figure caption). Why different terrain data (blanking or not) in bottom-left corner of the panels are shown?

(16) Fig.8a is slightly different from Fig.3b, why?

**Technical corrections**

1. Using "from" instead of the "of" in "temperature deviations (shading) within ±5 K of ERA5 reanalysis" may be better, in Line 235.

2. "evolution" in Line 270, and "East Asian Monsoon Index ()" in Line 271 can be removal with the EAMI defined previously.

3. "maintaining comparable spatial RMSE to observations" (in Lines 306-307) ? Is it "maintaining comparable spatial RMSE to CNN"?

---

## Author Comment (AC2)

**Response to Reviewers' Comments**

**Response to Reviewer #1:**

We thank this reviewer for the insightful comments and detailed suggestions on how to improve the manuscript. The manuscript has been further improved based on your comments. Below is the item-by-item reply to your questions and suggestions, the texts with normal font are your original comments, the texts with blue font are our responses and the texts with italics are the revised content of the manuscript.

Abstract:

The abstract is light on some specific details which would be valuable to a reader in understanding the contribution of the paper. While these may not be strictly required, adding them would significantly improve communication of the results.

Should mention the resolution of the GCM/ML, and that the GSRM simulation data is coarsened (coarsening is itself a difficult problem you are tackling here).

Reply: We appreciate the reviewer's comment regarding model resolution. In response, we have now explicitly specified the native spatial resolutions of the GSRM (5 km) and GCM (120 km) models in the revised abstract (L16-18).

*"This study develops a machine learning (ML)-based physics parameterization suite trained on 80-day global storm-resolving model (GSRM) simulation data (5km), attempting to replace all conventional physics tendencies in a general circulation model (GCM, 120km) for real-world simulations with realistic surface topography."*

(Optional) It is also worth highlighting the use of full topography.

Reply: Thank you for your comments. Done.

(Optional) What is the GCM used?

Reply: We have now added the specific model used in this study to the revised manuscript (L18-20).

*"The GSRM data are generated using the Global–Regional Integrated Forecast System (GRIST) and subsequently coarse-grained, after which the residual method is applied to derive the corresponding GCM physics tendencies."*

(Optional) Says comparison is being done against observations, it would be worth spelling out precipitation observations and historical reanalysis/ERA5. Currently reanalysis comparison is not mentioned in the abstract.

Reply: As suggested, the relevant content has been incorporated into the revised manuscript in L24-26.

*"It effectively mitigates the biases of excessively-strong rainbands and overly-wide width of the ITCZ in the conventional configuration, when compared with the Global Precipitation Measurement (GPM) data."*

**Overall comments:**

To evaluate the quality of the learning, the method output must be compared against the GSRM configuration of the model, which is congruent with the training procedure. Observations and long integrations may also be compared against, but it needs to be clear the extent to which observations are being matched by properly learning the training data, as opposed to just having less variability. This can only be done to the extent that the match of the GSRM with observations or reanalysis is known/presented. For example, does the GSRM have reduced precipitation compared to the CPS configuration? Does the mean precipitation of the long-term MPS configuration agree with the precipitation present in the GSRM training set? One would imagine it is not biased in the first few timesteps - does the precipitation produced by the MPS configuration stay consistent throughout the run, or does it reduce as the simulation leaves the initial condition?

Reply: Thank you for your comments. In response, we performed numerical experiments aligned with the GSRM simulation period using both the GRIST-CPS and GRIST-MPS configurations. Figure A1 shows the resulting tropical precipitation intensity–frequency distributions from GSRM, GRIST-CPS, and GRIST-MPS.

As shown in Figure A1a, the GSRM produces less total precipitation and less light rainfall compared to GRIST-CPS. The GRIST-MPS configuration effectively inherits these precipitation characteristics from the GSRM, also exhibiting reduced total and light precipitation relative to GRIST-CPS. Furthermore, comparisons between Figure A1a and A1b indicate that the precipitation distributions of GRIST-CPS and GRIST-MPS remain consistent across both the GSRM-aligned experiments and long-term climate simulations. This supports that GRIST-MPS retains the key precipitation features identified in the GSRM even under extended climate integration.

[Figure]

Figure A1. (a) The frequency probability distributions of tropical daily precipitation corresponding to the 80days-GSRM timeline obtained from GSRM (gray boxes), GRIST-CPS (yellow boxes) and GRIST-MPS (orange boxes). (b) As in (a) but for precipitation from 2001-2006 of GPM (gray boxes), 11 CMIP6 models ensemble mean (CMIP6-ENS; pink boxes), GRIST-CPS (blue boxes) and GRIST-MPS (orange boxes).

Regarding the potential long-term decline of precipitation in GRIST-MPS, an analysis of monthly tropical precipitation (Figure A2) reveals that GPM, GRIST-CPS, and GRIST-MPS all exhibit minimal long-time-scale drift. Importantly, there is no obvious trend in the precipitation of both GRIST-MPS and GRIST-CPS.

[Figure]

Figure A2. The time series of monthly tropical precipitation ($10°S − 10°N$) from 2001 to 2006. Black for GPM, blue for GRIST-CPS and red for GRIST-CPS.

The relevant content has been incorporated into the revised manuscript in L307-323.

"*The intensity–frequency distribution of precipitation reflects intrinsic model characteristics that remain stable over the course of a simulation. To assess whether the MPS faithfully captures the behavior of the GSRM, we conducted parallel experiments with the MPS and CPS using time periods aligned with the GSRM (i.e., the four cases listed in Table 2). Focusing on tropical precipitation (10°S–10°N), we categorize rainfall into four intensity ranges: light (0.1–10 mm day⁻¹), moderate (10–25 mm day⁻¹), heavy (25–50 mm day⁻¹), and extreme (>50 mm day⁻¹). As shown in Fig. 7a, relative to GRIST-CPS, the GSRM exhibits reduced total precipitation frequency and a lower frequency of light rainfall. GRIST-MPS consistently reproduces these features, with both total and light precipitation frequencies lower than in GRIST-CPS. Furthermore, comparing Figs. 7a and 7b reveals that both GRIST-CPS and GRIST-MPS display similar frequency characteristics in the GSRM-aligned experiments and the long-term free-run integrations, underscoring the robustness of these model behaviors.*

*Besides GPM observations, the ensemble means values of 11 CMIP6 models (CESM2, CESM2-WACCM, CMCC-CM2-SR5, E3SM-2-0, E3SM-2-0-NARRM, EC-Earth3, EC-Earth3-AerChem, GFDL-CM4, MRI-ESM2-0, SAM0-UNICON, TaiESM1; hereafter CMIP6-ENS) are included. In relative to GPM data, both CMIP6-ENS and GRIST-CPS overestimate total precipitation occurrence by 54% and 34%, respectively (Figure 7b)—consistent with earlier documented biases (Fu et al., 2024). The MPS reduces this discrepancy to 31%. It reduces light and heavy rain overprediction by 10% and 5%, respectively, while preserving observed extreme precipitation frequencies. This demonstrates that MPS effectively mitigates persistent precipitation distribution errors without compromising heavy-precipitation event statistics. Meanwhile, neither the CPS nor the MPS indicates a long-term artificial declining trend in precipitation (figure not shown).*"

Consider including a baseline similar to MPS but where the ML-based predictions of Q1 and Q2 are uniformly zero. At the least, the MPS model should better match the coarsened GSRM data than this baseline, and this comparison shows whether the MPS has any positive skill when integrated online (whether or not it outperforms CPS).

Reply: In our experimental framework, $Q_1$ and $Q_2$ represent the net effects of nearly all physical parameterizations. Setting them to zero would effectively deactivate the model's physics tendencies.

Such a configuration is unstable for real-world simulations, and is prone to fail under realistic conditions. Therefore, to fairly evaluate the MPS, we have designed experiments comparing it directly against the benchmark high-resolution model (GSRM) and the conventional physics scheme (CPS) on precipitation intensity–frequency distributions. The results (see the previous reply) demonstrate the superiority of the MPS, which shows a consistent improvement over the CPS. This confirms that the online application of the MPS provides tangible benefits to model performance.

The word "validation" only appears once in the manuscript, and it is not clear what validation dataset if any are used to measure overfitting and skill during training and hyperparameter optimization. On L91-92, it is implied that GSRM-style data is only used for training, and not for evaluation, making it hard to understand how out-of-sample performance on the loss could be evaluated. I am well aware of the high cost of GSRM output, but at least a few days of independent simulation data should be used for this purpose. Especially given the possibility of overfitting on a dataset only 80 days long, it is crucial to include validation metrics during training.

The implication of the lack of validation dataset is that the authors tuned the model on the measures being presented in the paper. If this is true, it significantly increases the potential any model skill is due to random chance, especially given that the final model was selected by testing an unnamed number of models.

Reply: We thank the reviewer for highlighting this issue. During the training process, we split the 80-day data into a training set and a validation set at a ratio of 7:1. Therefore, there should be no difficulty in evaluating the out-of-sample performance based on the loss, and relevant details have been added in the manuscript (L117–119) to explicitly clarify this sampling method, which was designed to robustly evaluate the model's out-of-sample performance and mitigate overfitting.

*"A 7:1 ratio was used to divide the dataset into training and validation sets. For each day, 12.5% of the time points were randomly allocated to the validation set, and the remaining 87.5% were used for training. This temporal sampling strategy supports a reliable assessment of the model's out-of-sample performance."*.

**Line:**

A few lines talk about scale-invariance, though none is shown here. L45: "Ideally, such models would not only perform robustly at a specific resolution but also enable a smooth transition across multiple meteorologically significant scales, from the typical GCM resolution (100 km) to the GSRM resolution (1 km)" and L119 "Although the present study coarse-grains GSRM data to a fixed resolution, the residual method allows efficient transitions 120 from arbitrarily high-resolution models to GCM target scales, thereby enabling the MPS to be inherently scale-aware." It is not clear to me here or from referring to Zhang and Chen 2016 how this is the case. For instance, a model which learns residuals at a given resolution cannot be applied zero-shot at other resolutions, any more than a model that learns the full field. The model does not appear to be scale-aware as mentioned on L120. I would suggest removing these references to smooth scale transitions, as they are not important to the primary contribution of this paper. If they are kept, they should be supported at least with a theoretical basis, if not experimental results. Note that the results here do successfully lead a coarser model to behave more like a finer model, but this is not what is generally meant by scale-awareness or by a smooth model transition across scales.

Reply: Thank you for this suggestion. We agree with the comment regarding the statement in Line 45, as the current study does not directly address GCM simulations at different resolutions, and such a claim cannot be substantiated here. Accordingly, we have removed that sentence from the manuscript.

We have retained the related methodological note in Line 119 and added further clarification. The
residual method provides a flexible framework for constructing multi-scale training datasets, which is
a key step toward achieving scale-aware modeling in future work. Given its conceptual importance,
we have preserved and expanded on this point in L128-133 of the revised text.:

*"While the present study coarse-grains GSRM data to a fixed resolution, the residual method is*
*inherently adaptable. It can seamlessly bridge models of arbitrarily high resolution to GCM target*
*scales. Establishing a robust physical correspondence between GSRMs and GCMs can not only allow*
*GCMs to mimic certain behaviors of GSRMs, but also opens the door to unified simulations of*
*atmospheric processes within a single modeling framework—enhancing both theoretical*
*understanding and predictive skill across multiple timescales."*

There are some conflicting statements about the configuration of the GCM - it's an important detail
whether it's using explicit convection or a cumulus scheme. On L93-94 it says the cumulus scheme is
disabled, but on L94 the GSRM is compared against the GCM with a conventional cumulus
parameterization. On L103 it mentions the cumulus parameterization being turned on. Any results of
MPS should be compared in skill against the optimal GCM configuration for the base model, in this
case it seems one with the conventional cumulus parameterization turned on, even if MPS is hooked
into a version of the GCM without cumulus parameterization. I think that's what you're doing, but
perhaps L93-94 should be modified.

Reply: We thank the reviewer for this comment. The text in the original Lines 93–94 has been revised
to eliminate ambiguity and is now located in Lines 99–100 of the updated manuscript.

*"The GSRM setup uses the nonhydrostatic dynamical core with explicit convection, in which*
*cumulus scheme is disabled, following the approach of Zhang et al. (2022)."*

L117-118: How are Q1 and Q2 computed numerically? For example, are the terms computed by taking
gradients of the coarsened fields according to a given numerical scheme (if so, something like "where
gradients are computed using [... ]"), or by taking the tendency of the dynamical core in coarse mode?

Reply: Thank you for your comment. The gradient term is calculated using the center difference
method. This clarification has been added to the manuscript in L127-128:

*"Specifically, the gradient operator is achieved via the center difference method applied to the*
*coarse-grained fields, following the governing equations shown in Figure 1 (the middle section of the*
*left panel)."*

L142: Please clarify whether model outputs are referring to GSRM values or ML outputs.

Reply: The term "model outputs" here refers to the data produced by the coarsened GSRM and we
have clarified this definition in the revised manuscript.

"column" only appears as of line 146, and I only realized this is a column model by line 156. The
model being column-local is noteworthy and should be mentioned much earlier.

Reply: We appreciate the suggestion. The term "column" is now introduced earlier in the manuscript
(L77) to familiarize readers with the concept before it is used in subsequent sections.

L152: What optimizer is used?

Reply: The use of the Adam optimizer has been specified in the revised manuscript.

L152: How was model skill evaluated when performing hyperparameter optimization?

Reply: During hyperparameter tuning, the mean absolute error (MAE) was employed as the evaluation
metric. This information has been specified in the revised manuscript (L166–167).

*"The mean absolute error (MAE) loss was selected over the mean squared error (MSE) loss as*
*the loss function, as it demonstrated superior performance during initial training phases."*

L153: The stated decay rate of 1e-6 per epoch is either incredibly large (if directly multiplicative,
giving a final epoch LR of (3e-4)*(1e-600), or incredibly small (if the multiplier is (1 - rate), giving a
final epoch LR of 2.9997e-4). Is this accurate?
Reply: We are grateful to the reviewer for highlighting this inaccuracy. The initial mention of a
"decaying learning rate" was incorrect; the model was in fact trained using a constant learning rate of
$3\times10^{-4}$, with a per-epoch weight decay of $10^{-6}$ applied within the Adam optimizer. This has been
clarified in L165–167 of the revised manuscript, which now explicitly states:
*"We used the Adam optimizer with a constant learning rate of $3\times10^{-4}$ and a weight decay of $10^{-6}$*
*per epoch. The mean absolute error (MAE) loss was selected over the mean squared error (MSE) loss*
*as the loss function, as it demonstrated superior performance during initial training phases."*
L153-154: it is conventional to say you are using a MAE loss function, and how the values are
normalized when computing the loss (in this case, it would appear to be using the min-max scaling
mentioned elsewhere). It could be worth remarking on why you chose MAE loss over MSE, if there is
a reason.
Reply: We selected the mean absolute error (MAE) as the loss function based on its superior
performance over the mean squared error (MSE) in preliminary tests. This choice, made during initial
model development, was consistently applied in all subsequent experiments. The rationale has been
clarified in the manuscript (L166–167).
*"The mean absolute error (MAE) loss was selected over the mean squared error (MSE) loss as*
*the loss function, as it demonstrated superior performance during initial training phases."*
L160: Please mention the batch size here alongside the number of samples per epoch.
Reply: The batch size used in our experiments is 1024, and this information has been added to the
manuscript.
L162-172: You mention a process of training many models and evaluating for the best candidates.
Please specify how many models were trained for this selection process.
Reply: The model was trained across eight random seeds, from which the one achieving the best
performance was selected for final analysis. This information has been added to the manuscript (L185–
187).
*"For the NNs that meet the above criteria, we will continue their integration until collapse.*
*Among the 8 NNs, 2 NNs can integrate stably for more than 6 years, and we have selected the one with*
*better performance as the optimal MPS."*
L171: Please specify exactly what is meant by "demonstrating stability in online integration". For
example, was there a pass/fail measure of stability? Does the paper present the first trained model for
which this stability measure was passed?
Reply: The selection process consisted of two stages. First, we identified seeds capable of maintaining
stable online integration for at least three months. Four out of eight initial seeds met this criterion.
These were integrated further until simulation termination. From the two seeds that remained stable
for over six years, we selected the one that produced the most realistic precipitation pattern as the final
model. This description has been added to Lines 183–187:
*"The final selection of our optimal MPS is based on a dual evaluation: satisfying offline*
*performance benchmarks and demonstrating stability in online integration which must maintain stable*
*online integration for more than 3 months. For the NNs that meet the above criteria, we will continue*
*their integration until collapse. Among the 8 NNs, 2 NNs can integrate stably for more than 6 years,*
*and we have selected the one with better performance as the optimal MPS."*

L173-183: No edit needed here, but this result is quite curious and I am skeptical of the interpretation,
since the linear interpolation doesn't grant any true additional temporal resolution, and emulator
models successfully operate at 6-hour timesteps. It may be that the model behaves better online when
it must make 3 of roughly the same (residual) prediction, and its errors get averaged or the intermediate
values serve as a type of data augmentation. You might consider for your own purposes training this
model with a 1-hour timestep but linearly interpolating the training data from 3-hour intervals down
to 1-hour, and see if you get a similarly stable model. This would prove out that it is in fact the 20-
minute temporal resolution which is key, and not the structure of linearly-interpolated data.

Reply: Yes, we also agree that a 20-minute temporal resolution is essential. As in the title of Section
2.4 "Importance of using balanced spatiotemporal sample and temporal resolution alignment". The
current 20-minute interval is chosen to align with the physical timestep of the GCM experiment,
thereby ensuring smoother coupling within the dynamical framework and enhancing the overall
integration quality.

The use of linearly interpolated data can be regarded as a simplified, or "poor man's" approach to
finely sampling the model state, which would otherwise be more expensive in terms of storage for
production runs. The original phrasing may be a little bit confusing, and we have now clarified and
improved it accordingly (L205-209).

*"Linear interpolation is not the only means of generating more data; one may alternatively choose*
*to directly sample the model state at finer, aligned timesteps, but this is more expensive. While temporal*
*resolution alignment is the key, our results demonstrate that the linear interpolation of large-scale*
*state variables serves as an effective ecomonical alternative to finely sampled model output."*

L186-188: It is not right to say the effective number of training samples is tripled, given these are not
independent samples but rather completely linearly dependent on existing training samples. Rather,
this  is  similar  to  other  regularization  strategies  used  for  improving  stability,  e.g.
https://arxiv.org/abs/1912.02781 or Bishop (1995) "Training with Noise is Equivalent to Tikhonov
Regularization"

Reply: We thank the reviewer for pointing this out. This inaccurate statement regarding the effective
sample size has been corrected in the revised manuscript (L200-202).

*"First, augmenting the dataset by a factor of three provides a regularization effect, which is*
*known to improve model stability and generalization (Bishop et al. 1995)."*

L215-216: It is somewhat problematic to compute precipitation in this way, as the value is really
precipitation minus evaporation. This significantly affects the headline result of the paper (improved
precipitation skill through lower RMSE). It would be appropriate to compare against precipitation
minus evaporation for the CPS configuration, and to appropriately describe these values in later
sections. For example, this significantly affects the interpretation of Figure 7.

Reply: We sincerely thank the reviewer for raising this important issue. To address it, we have revised
the MPS precipitation diagnosis to include the surface evaporation term, as now shown in L233–236
of the manuscript. This addition, formulated as *"Surface precipitation (Prec; unit: kg.m-2.s-1 or mm.s-*
*1) is diagnosed by the MPS via vertically integrated moisture tendency equation plus the evaporation*
*(Evap), calculated using:  $Prec = -\frac{1}{g} \int (Q_2/L)dp + Evap$. The evaporation term is included solely*
*for diagnostic purposes; the precipitation input provided to the land surface model actually excludes*
*this term, as a tuning procedure."* This allows a consistent and physically more complete comparison
with CPS and GPM outputs. While the inclusion of evaporation modifies certain details in the spatial distribution and frequency figures of precipitation, our key conclusions remain robust. All related
figures (Figures 3, 4, 7, 9, 10, and 11) have been thoroughly revised to incorporate this update.
L223-225: Please quantify the reduction in computational cost of GRIST-MPS (using CPS for the
layers described) compared with GRIST-CPS.
Reply: The computational efficiency of the baseline GRIST-MPS at 120 km is nearly comparable to
that of the CPS at the relatively coarse resolution used in our main experiments. This is because the
computational cost of the CPS is already moderate in the 120 km model configuration. This contrasts
with previous AI-physics studies employing superparameterized models, in which the physics suite
(mainly 2D cloud resolving models) constitutes a substantial portion of the overall model workload.
However, in high-resolution tests emphasizing scalability, the optimized version of GRIST-MPS
achieves over 30% improvement in computational efficiency, showing great potential. This
explanation has been added to the manuscript (L246–249).
*"Furthermore, when extended to higher resolutions, it reduced computational costs by over 30%*
*with a similar configuration (Duan et al., 2025) when optimizations have been carried out. This is*
*attributed to the more optimizable computational structures of ML models (convolution, matrix*
*multiplication), while the conventional physics schemes hardly support these optimization potentials"*
L228: What years are used for the AMIP-style simulations? Are these related to the years used for the
GSRM training simulations?
Reply: It is from 2001-2006, determined by the use of sea-surface temperature and sea ice
concentration. This period covers a part of the GSRM simulation: April 1, 2005 – April 20, 2005. This
date was touched by the model only after it had already been integreated for nearly 5 years. We believe
that this had no impact on the experiment. The initial year has been added in the manuscript.
L243-247: Please comment on the relative strength of the ITCZ between the models, and quantify the
underestimation and overestimation of precipitation in the southern oceans.
Reply: We appreciate the reviewer's suggestion to include more quantitative analysis. In response, we
have conducted and now provide quantitative assessments of ITCZ properties (**intensity and width**)
and southern ocean precipitation in the revised manuscript (L267-L272), further supporting the
conclusions drawn.
*"During JJA months (Table 4), GRIST-MPS produces a more realistic ITCZ in terms of both*
*strength and width, despite exhibiting a slightly lower pattern correlation coefficient (PCC: 0.86) than*
*GRIST-CPS (0.94). Following established practice, we quantify ITCZ strength by the maximum zonal-*
*mean tropical precipitation rate (Wodzicki and Rapp, 2016) and define its width as the tropical*
*latitudinal extent with precipitation exceeding 5 mm day⁻¹ (Byrne et al., 2018). The MPS accurately*
*captures the ITCZ strength (8.69 mm day⁻¹), closely matching the GPM estimate (8.54 mm day⁻¹). By*
*contrast, the CPS produces an excessively strong (10.09 mm day⁻¹) and overly broad (9.60°) rain band,*
*compared with the observed width of 7.44° and the MPS width of 7.76°."*
L249-251: Can you give evidence this reduced RMSE is due to improved placement of the ITCZ and
not reduced strength, for example by using a measure that accounts for the variance of the model?
Reply: As indicated in our previous response, while the ITCZ intensity in GRIST-MPS is indeed
weaker than that in the CPS, it aligns more closely with GPM observations. In addition, the simulated
ITCZ width in GRIST-MPS is also quantitatively closer to observations. We believe that these two key
metrics already demonstrate the improvement of GRIST-MPS and its better consistency with reality.
Please refer to the previously provided results for details.
Figure 4: GRIST-MPS seems to have significantly reduced mean precipitation, but it is hard to tell from color maps alone. Please include a measure of mean bias, as well as a measure of skill which does not improve for models with less variability, such as the Pearson correlation coefficient. This would make it clear the MPS model is outperforming the CPS model.

Reply: We have updated the evaluation to include the spatial pattern correlation coefficient. As shown in Table 4, GRIST-MPS performs slightly less well than GRIST-CPS in terms of spatial distribution similarity. Nevertheless, subsequent analyses demonstrate that GRIST-MPS retains clear advantages over GRIST-CPS in simulating key features such as the ITCZ and Southern Ocean precipitation. It should also be noted that there remains potential for further improvement in spatial distribution—since the current model was trained on only 80 days of data, we anticipate that increasing the training dataset will lead to better spatial performance in the future. The corresponding descriptions have been revised in L267–268.

*"GRIST-MPS produces a more realistic ITCZ in terms of both strength and width, despite exhibiting a slightly lower pattern correlation coefficient (PCC: 0.86) than GRIST-CPS (0.94)."*

Figure 5: The rainbow color bar is not doing you any favors here, making it look like you have a drastic reduction in precipitation. Using a more linear color bar would give a more accurate representation of the data.

Reply: We appreciate the suggestion and have implemented a linear color bar in the revised figures.

L259-260: Does this really show superior constraint of the ITCZ width with MPS? The green contour lines appear similar to CPS, and past that the ITCZ appears weakened more than narrowed. The lack of a maximum around -15S Jan-March for MPS is also notable, weakened variability in this region is also apparent in Figure 4.

Reply: Thank you for these insightful comments. **We have quantified the ITCZ width and intensity using established metrics, and the results remain consistent with our main conclusions.** We agree that the seasonal cycle of precipitation alone does not fully capture ITCZ variability. Regarding the precipitation center near 15°S, this feature is indeed influenced by the evaporation term, which was not fully considered in our initial diagnostic analysis. After incorporating evaporation, the precipitation center in this region from January to March is now clearly represented. The relevant descriptions have been revised in the manuscript (L284–288).

*"Both configurations accurately reproduce the observed seasonal migration of tropical precipitation maxima (Figure 5), with boreal summer peaks centered near 5-10°N aligned with the northward-migrating ITCZ. However, systematic discrepancies emerge in the meridional range of precipitation representation: GRIST-CPS overestimates the central precipitation intensity, generating strengthened rainfall distributions of overactive convective initiation in cumulus parameterizations. GRIST-MPS demonstrates slightly overestimation of the precipitation range throughout the seasonal cycle."*

Figure 7: Please include the coarsened GRIST data used as target for the MPS model. It is not clear whether the MPS is doing "better" because it is accurately representing reduced precipitation as seen in the GSRM, or because it has weakened variability in general compared to CPS.

Reply: Thank you for this suggestion. As shown in Figure A1, we conducted an 80-day experiment aligned with the GSRM period. The results indicate that GRIST-MPS is closely aligned with the GSRM in precipitation characteristics, and this consistency is maintained in long-term climate integration experiments (Figure 7). These findings demonstrate that GRIST-MPS successfully inherits the precipitation behavior of the high-resolution model and outperforms GRIST-CPS in capturing these features.

L309-310: Please quantify improved spatial pattern alignment. RMSE is not sufficient for this.

Reply: Thank you for this suggestion. We have provided a new Table 4, including spatial pattern correlation, to reflect this point.

**Response to Reviewer #2:**

We thank this reviewer for the insightful comments and detailed suggestions on how to improve the manuscript. The manuscript has been further improved based on your comments. Below is the item-by-item reply to your questions and suggestions, the texts with normal font are your original comments, the texts with blue font are our responses and the texts with italics are the revised content of the manuscript.

**General comments**

This preprint describes the improved long-term global climate simulations using a general circulation dynamic core coupled with machine-learning (ML)-based physics instead of the conventional parameterization schemes. While economic ML large models have been long doubted for lacking of physical constraints, the dynamical models suffer from expensive computational cost with conventional parameterization or unstable computation with online ML physics schemes (MPS). The present study shows a practical way of coupling the ML-physical processes for stable model integration by selecting balanced spatiotemporal resolution of the training data, besides the evaluation of three neural network architectures. The proposed method effectively ameliorates the precipitation simulation and structure of thermodynamic quantities (T and q) via the improved convective-diabatic interaction, even though no physical feedback to momentum is considered. Descriptions of the methodology and corresponding consummate skill in the MPS implementation, which must be interested and also helpful to researchers in this field, are provided in detail for reference.

Reply: We sincerely appreciate your positive feedback and valuable comments on our work.

The use of high-resolution model output as the training data, being of a result of multi-scale dynamical interaction, helps the neural network to learn detailed spatial structure and temporal variation of the corresponding variables from the fine-modelled data. The induced MPS is then capable of reproducing the high-resolution-model pattern of precipitation and prognostic variables with coarse-resolution GCM, just as show in this preprint.

Beside the quality improvement, the computational cost is also an issue of concern. It is better to add a short note on the cost of MPS vs CPS. In this study, the hybrid coupling of CPS and MPS to dynamical core, in addition to retaining of radiation scheme for the land surface model, seems introduce excessive computation. How about the cost reduction with the MPS?

Reply: The computational cost of the MPS in our AMIP experiments is nearly equivalent to that of the CPS. Two factors contribute to this: the relatively low weight of physical parameterization in the total computational budget with this configuration, and the fact that other physical process diagnostics (besides convection) are also enabled. It is important to note that in high-resolution experiments using an optimized version of the MPS under similar configurations (Duan et al., 2025), computational efficiency improves by more than 30%. This clarification has been incorporated into the manuscript (Lines 245–249).

*"This hybrid replacement strategy demonstrates that partial physics‒ML integration can achieve climate fidelity comparable to a full replacement and mitigating numerical instability. Furthermore, when extended to higher resolutions, it reduced computational costs by over 30% with a similar configuration (Duan et al., 2025) when optimizations have been carried out. This is attributed to the more optimizable computational structures of ML models (convolution, matrix multiplication), which are clearly difficult to achieve in conventional schemes."*

Another concern issue is the diagnosis of the $Q_1$ and $Q_2$ on pressure levels. The interpolation between the model levels and the pressure levels introduces errors inevitably. Why not diagnose directly on the model levels?

Reply: Thanks for pointing out this issue. We assume the interpolation errors are overall small in the
entire workflow. The reasoning is further elaborated in the manuscript (Lines 143–154).

*"**Vertical coordinate alignment.** For machine learning training, it is more desirable to use the*
*model's native hybrid coordinate, which avoids topographic distortion during runtime. Calculating*
*$Q_1/Q_2$ in the residual method requires first obtaining the advection tendencies. However, directly*
*computing advection tendencies offline on the hybrid vertical coordinate is inaccurate because the*
*generalized vertical velocity cannot be reliably reconstructed from coarse-grained data. It would*
*require the generalized vertical velocity to be explicitly saved during the online integration, which is*
*currently not available. Meanwhile, in this study, we prefer to confine our training workflow to*
*standardized pressure-level variables as inputs, ensuring that the workflow has the potential to be*
*consistently applicable to non-GRIST GSRM datasets.*

*To reconcile this discrepancy, we implement a two-step procedure. In Step I, GSRM variables on*
*the hybrid coordinate are interpolated to pressure levels for the sole purpose of computing advection*
*tendencies. In Step II, the resulting advection tendencies are interpolated back to the model's hybrid*
*coordinate, where $Q_1$ and $Q_2$ are then derived. Ultimately, all training inputs (U, V, T, q, P) and*
*outputs ($Q_1$ and $Q_2$) are defined on the model's hybrid vertical coordinate, ensuring compatibility*
*with the runtime model structure while preserving physical accuracy in the derivation process."*

The preprint is generally well written and organized. The technical details must be highly
evaluated for the online stable integration. I recommend accept for publication after necessary
revisions.

**Specific comments**

(1) The GRIST model should be mentioned in the abstract.

Reply: We thank the reviewer for this suggestion. The GRIST model has been explicitly mentioned in
the abstract of the revised manuscript.

(2) Please confirm the "precipitation frequency-intensity spectra" in Line 23, may be "precipitation
frequency" because no frequency-intensity spectra is shown in the preprint.

Reply: As suggested, we have revised "precipitation frequency-intensity spectra" to "precipitation
frequency" in Lines 26–27 and L351–353.:

*"Moreover, the hybrid ML-GCM better captures precipitation frequency, notably mitigating the*
*overproduction of light tropical rainfall and improving the simulation of heavy rain rates."*

*"Key improvements include more accurate ITCZ strength and width, phase-aligned midlatitude*
*storm tracks, and improved precipitation frequency, particularly the improved light (0.1-10 mm/day)*
*and heavy range (25-50 mm/day)."*

(3) Please rephrase the "explicitly resolved global storm-resolving models" in Line32

Reply: We thank the reviewer for this suggestion. Done.

(4) Line 90, please note the abbreviation "PhysW" here.

Reply: We thank the reviewer for this suggestion. The corresponding abbreviation has now been
defined in the revised manuscript (L95-96).

*"For this study, we adopt the weather physics (PhysW) suite of the model as the basis of model*
*development (see Li et al. 2023 for details)."*

(5) (Line 120) In case of a different resolution, the use of MPS needs additional training or not? For
the specific resolution, the trained MPS may resolve interactions between systems above the doubled
grid scale. For finer resolution, however, the trained MPS may miss the processes smaller than training data resolution (0.25 degree). So, is it really scale-aware?

Reply: Yes. Applying the MPS trained at 0.25-degree to a substantially finer resolution (e.g., 5km)
would require it to be systematically retrained. Regarding scale adaptability, we have revised the text
(Line 128–132) to more accurately emphasize the inherent advantage of the residual method:

*"While the present study coarse-grains the GSRM data to a fixed resolution, the residual method*
*itself is inherently scale-adaptive. It can seamlessly bridge models of arbitrarily high resolution to a*
*target coarse scale. This property potentially allows the MPS to be intrinsically scale-aware, enabling*
*a flexible construction of the training datasets across different resolutions."*

(6) Line 124, is the "GSRM" a typo of "GCM"?

Reply: We thank the reviewer for noting this. The term should be "GSRM" here. We have revised the
text in L133-135 to clarify that GSRM simulations from various institutions can universally serve as
training data sources, thereby improving the clarity of this point.

*"(i) ability to enable the transfer of scale interactions represented in GSRMs to a target GCM*
*resolution"*

(7) Why not directly output (U, V, T, q, P) data in a 20-minute interval in the GSRM simulation (Lines
142 & 179)? Only due to the limitation of data storage?

Reply: Yes. If sufficient resources are available to generate more frequent data outputs aligned with
the GCM timestep, we recommend doing so, in which case the linear-in-time interpolation described
above can be omitted.

Meanwhile, given that storing additional GSRM data is highly expensive, achieving satisfactory
quality with less frequent outputs would be of considerable practical value. Such experience could also
provide useful guidance for the design of future studies.

(8) Please check the formula of *Prec* in Line 216. What unit is used here?

Reply: We thank the reviewer for highlighting this issue. The unit of $kg.m^{-2}.s^{-1}$ is derived from the
expression: $-\frac{1}{g}\int (Q_2/L)dp$. Furthermore, in accordance with the reviewer 1's suggestion, the
contribution of evaporation has now been incorporated into the precipitation diagnosis. The revised
text in Lines 233–235 now reads:

*"Surface precipitation (Prec; unit: $kg.m^{-2}.s^{-1}$ or $mm.s^{-1}$) is diagnosed by the MPS via vertically*
*integrated moisture tendency equation plus the evaporation (Evap), calculated using: Prec =*
$-\frac{1}{g}\int (Q_2/L)dp + Evap."$

Also please note that all the figures adopt the unit of "mm/day" for precipitation.

(9) Please define X, μ and σ in Line 269.

Reply: We thank the reviewer for this suggestion. The descriptions of the aforementioned symbols
have been added to the manuscript in L297–298.

*"where $X$ is the corresponding variable $(U, SLP)$, where $\mu$ represents the mean of variable $X$*
*and $\sigma$ represents the standard deviation of variable $X$."*

(10) The word "extreme event" is better replaced with "heavy-precipitation event" for easy
understanding. The extreme event, out of the training data, is difficult for MPS to resolve.

Reply: We agree with the suggestion and have replaced "extreme event" with "heavy-precipitation
event" in the revised manuscript.

(11) What is the "weight file" in Line 292.

Reply: We appreciate the reviewer's query. The "weight file" is file storing the trained parameters (weights) and the neural network structure. We have added an explanatory note in Lines 327–328 to ensure proper interpretation of this component.

*"The switch of each network during the GRIST-MPS runtime only needs to change the NN file which contains the weights and structures of each NN."*

(12) The sentence on Line 322 needs to be revised. Do you mean: CNN nearly doubled the frequency of light precipitation occurrence in conventional GCMs?

Reply: We thank the reviewer for this comment. The suggested revision has been implemented in the revised manuscript.

(13) Can the GRIST-MPS stably run more than 6 years?

Reply: Yes, it can. The stable integration time depends on tuning. The current integration length of the MPS used for this paper is approximately 6 years and 1 month. This is determined based on a balance between simulation accuracy and long-term stability. A further extension of this stable integration time and a more accurate version is a focus of our ongoing development.

(14) It is helpful to label the averaged pressure in right side of panels in Fig.2

Reply: Figure 2 is presented on the model's native hybrid coordinate levels. Because this coordinate system varies with topography, the pressure at a given level is not constant, and a precisely averaged pressure is not defined. Therefore, the mean pressure is omitted.

(15) In Fig.3, temperature difference should be the shaded (being wrongly noted in figure caption). Why different terrain data (blanking or not) in bottom-left corner of the panels are shown?

Reply: The figure caption has been revised accordingly. This occurs because the zonally averaged data will exhibit a "missing-value" pattern once a single missing-value point is introduced by the topography. The terrain differences observed here arise from the interpolation from model levels to pressure levels, together with minor discrepancies in surface pressure between the CPS and MPS simulations.

(16) Fig.8a is slightly different from Fig.3b, why?

Reply: Thanks for your sharp observation, helping us identify an inconsistency in the original Figure 8, where only the final 5 years of data were plotted. To ensure consistency across all analyses, we have updated Figure 8 to use the same data period (2001-2006) as Figure 3, now reflected in the revised manuscript.

**Technical corrections**

1. Using "from" instead of the "of" in "temperature deviations (shading) within $\pm 5$ K of ERA5 reanalysis" may be better, in Line 235.

Reply: Revised.

2. "evolution" in Line 270, and "East Asian Monsoon Index ()" in Line 271 can be removal with the EAMI defined previously.

Reply: Revised.

3. "maintaining comparable spatial RMSE to observations" (in Lines 306-307) ? Is it "maintaining comparable spatial RMSE to CNN"?

Reply: Yes. Revised.